# Quantification of anthropogenic and marine sources to atmospheric

1

2

# mercury over the marginal seas of China and impact on the sea-air

| 3  | exchange of mercury                                                                                                                                                                    |
|----|----------------------------------------------------------------------------------------------------------------------------------------------------------------------------------------|
| 4  |                                                                                                                                                                                        |
| 5  | Xiaofei Qin <sup>1</sup> , Hao Li <sup>2</sup> , Jia Chen <sup>3</sup> , Junjie Wei <sup>4,5</sup> , Hao Ding <sup>4,5</sup> , Xiaohao Wang <sup>3</sup> , Guochen Wang <sup>2</sup> , |
| 6  | Chengfeng Liu², Da Lu², Shengqian Zhou², Haowen Li², Yucheng Zhu², Ziwei Liu², Qingyan Fu <sup>6</sup> ,                                                                               |
| 7  | Juntao Huo <sup>3</sup> , Yanfen Lin <sup>3</sup> , Congrui Deng <sup>2</sup> , Yisheng Zhang <sup>1*</sup> , Kan Huang <sup>2, 7, 8,9*</sup>                                          |
| 8  |                                                                                                                                                                                        |
| 9  | <sup>1</sup> School of Environmental and Municipal Engineering, Qingdao University of Technology, Qingdao,                                                                             |
| 10 | China                                                                                                                                                                                  |
| 11 | <sup>2</sup> Shanghai Key Laboratory of Atmospheric Particle Pollution and Prevention (LAP <sup>3</sup> ), National                                                                    |
| 12 | Observations and Research Station for Wetland Ecosystems of the Yangtze Estuary, Department of                                                                                         |
| 13 | Environmental Science and Engineering, Fudan University, Shanghai, China                                                                                                               |
| 14 | <sup>3</sup> Shanghai Environment Monitoring Center, Shanghai, China                                                                                                                   |
| 15 | <sup>4</sup> Key Laboratory of Environmental Pollution Control Technology of Zhejiang Province, Hangzhou,                                                                              |
| 16 | Zhejiang, China                                                                                                                                                                        |
| 17 | <sup>5</sup> Environmental Science Research & Design Institute of Zhejiang Province, Hangzhou, Zhejiang,                                                                               |
| 18 | China                                                                                                                                                                                  |
| 19 | <sup>6</sup> Shanghai Academy of Environmental Sciences, Shanghai, China                                                                                                               |
| 20 | <sup>7</sup> Institute of Eco-Chongming, Shanghai, China                                                                                                                               |
| 21 | <sup>8</sup> IRDR ICoE on Risk Interconnectivity and Governance on Weather/Climate Extremes Impact and                                                                                 |
| 22 | Public Health, Fudan University, Shanghai, China                                                                                                                                       |
| 23 | <sup>9</sup> Observation and Research Station of Huaniaoshan East China Sea Ocean-Atmosphere Integrated                                                                                |
| 24 | Ecosystem, Ministry of Natural Resources, Shanghai, China                                                                                                                              |
| 25 | Correspondence: huangkan@fudan.edu.cn (Kan Huang); doctorzys@163.com (Yisheng Zhang)                                                                                                   |
| 26 |                                                                                                                                                                                        |
| 27 | Abstract                                                                                                                                                                               |
| 28 | Mercury in the atmosphere is a crucial environmental concern due to its toxicity and ability to travel                                                                                 |

long distances. In the marginal seas, the contributions of terrestrial anthropogenic vs. natural sources on atmospheric mercury have been rarely quantified and their roles in mercury sea-air exchange are not well understood. To address this issue, this study integrated observations from island, cruise, and inland campaigns. The mean concentrations of total gaseous mercury (TGM) were  $2.32 \pm 1.02$ ng/m³ (Bohai Sea),  $2.55 \pm 0.55$  ng/m³ (Yellow Sea), and  $2.31 \pm 0.81$  ng/m³ (East China Sea), respectively, with coastal regions exhibiting significantly higher values than open ocean areas due to continental outflows. Positive correlations were observed between TGM concentrations and environmental parameters such as temperature, relative humidity, and wind speed, indicating the significant influence of natural sources on atmospheric mercury in the marine environment. By utilizing a receptor model and linear regression analysis, a robust method was developed to quantitatively estimate the contribution of anthropogenic and natural sources to TGM. Anthropogenic sources accounted for an average of 59%, 40%, and 27% of TGM over the Bohai Sea, Yellow Sea, and East China Sea, respectively. The sea-air exchange fluxes of mercury were estimated as  $0.17\pm0.38$ ,  $1.10\pm1.34$ , and  $3.44\pm3.24$  ng m<sup>-2</sup> h<sup>-1</sup> over the three seas above, respectively. Stronger anthropogenic mercury emissions in the northern China partially explained the suppressed sea-air exchange fluxes of mercury in the Bohai Sea. This study elucidated the role of anthropogenic emissions in shaping the marine atmospheric mercury and the sea-air exchange fluxes, thereby informing valuable assessments regarding the influence of future reduced anthropogenic mercury emissions on the marine mercury cycle and ecosystems.

### 1. Introduction

Mercury is a ubiquitous toxic pollutant that can cycle among atmospheric, aquatic, and terrestrial environments (Mason et al., 2012; Lamborg et al., 2014). Anthropogenic discharge of mercury can be transported into marine atmospheres, subsequently entering oceans via wet and dry depositions, constituting a primary source of marine mercury (Outridge et al., 2018). A fraction of mercury that enters oceans can undergo methylation and bioaccumulate in the food chain, thereby posing health risks to humans through the consumption of methylmercury-contaminated seafood; another fraction converts to dissolved gaseous mercury and can escape from surface seawater through sea-air exchange processes (Lavoie et al., 2018; Obrist et al., 2018). This sea-air exchange

is pivotal to the biogeochemical cycling of mercury, as it influences mercury concentrations in seawater, oceanic mercury accumulation rates, and methylmercury production (Mason et al., 2017; Ci et al., 2016). Simultaneously, the sea-air exchange of mercury represented the largest flux between different environmental media within the global mercury cycle. Previous estimates indicated that the release of gaseous elemental mercury from the global ocean contributed approximately one-third of the global atmospheric mercury emissions (Horowitz et al., 2017).

Numerous studies have emphasized the impact of anthropogenic sources on marine atmospheric mercury. For instance, one study conducted over the Bohai Sea revealed that the increased concentration of gaseous elemental mercury (GEM) resulted from the long-range transport of mercury released from anthropogenic sources (Wang et al., 2020). An island investigation over the East China Sea showed the outflow from mainland China was the primary contributor to atmospheric GEM (Fu et al., 2018). Cruises campaigns over the East China Sea and South China Sea observed elevated GEM concentrations at sites proximate to mainland China, indicating the prominent influence of terrestrial emissions (Fu et al., 2010; Wang et al., 2016a). Additionally, studies in the Gulf of Mexico, North Atlantic Ocean, and Mediterranean Sea also attributed significant portions of atmospheric mercury to anthropogenic emissions (Obrist et al., 2018). Although isotopic signatures have been widely applied to source apportionment of atmospheric mercury, current isotopic methods still exhibit significant uncertainties due to the poor understanding of isotopic compositions of gaseous elemental mercury emitted from various sources and fractionation processes of Hg isotopes during atmospheric transformations (Fu et al., 2018). Additionally, this approach requires specialized isotopic measurements unavailable for routine monitoring. At present, quantitative analyses of anthropogenic contributions to marine atmospheric mercury remain limited.. Although annual global anthropogenic atmospheric mercury emissions have been approximated to reach 2300 tons, accounting for about one-third of global atmospheric mercury emission (Pirrone et al., 2010; Zhang et al., 2016), the specific contributions to marine atmospheric mercury remained poorly delineated, thereby constraining insights into the oceanic mercury cycling dynamics. In this regard, it is imperative to develop methodologies capable of quantifying the contributions from anthropogenic sources to marine atmospheric mercury, particularly in critical marginal seas, which served as essential biogeochemical interfaces between

landmasses and open oceans. Previous studies have indicated that the importance of the mercury cycling in offshore ecosystems approximated that within open oceanic environments (Fitzgerald et al., 2007). Marginal seas functioned not only as natural sinks for terrestrial mercury but also represented significant sources of atmospheric mercury (Ci et al., 2011). Given that China ranks as the foremost global emitter of anthropogenic atmospheric mercury (Pacyna et al., 2016; Pacyna et al., 2010; Zhang et al., 2015), its emissions inevitably exert profound influences on adjacent marginal seas.

Anthropogenic inputs influenced not only the concentrations of atmospheric mercury but also the dynamics of mercury sea-air exchange. Given that Hg<sup>0</sup> in the surface oceanic waters frequently exceeded its saturation levels, the prevailing direction of sea-air exchange was predominantly upward, facilitating the efflux of mercury from the ocean to the atmosphere (Andersson et al., 2008b; Mason et al., 2001; Huang and Zhang, 2021). The sea-air exchange of Hg<sup>0</sup> was governed by the concentration gradients at the atmosphere-seawater interface (Soerensen et al., 2013), which were influenced by the spectrum of physical and chemical processes within seawater, as well as meteorological conditions and ambient GEM concentrations (Costa and Liss, 1999; Mason, 2009; Selin, 2009). Previous studies illuminated the direct impact of dissolved gaseous mercury (DGM) in surface waters on Hg<sup>0</sup> fluxes, while photochemical reduction of Hg (II) has been identified as the principal mechanism driving the DGM generation in marine settings (Amyot et al., 1994; Huang and Zhang, 2021). Field measurements observed nocturnal peaks in DGM and Hg<sup>0</sup> fluxes, implying that dark reduction processes may significantly contribute to these dynamics (O'driscoll et al., 2003; Fu et al., 2013). Hg<sup>0</sup> fluxes increased 2-4 folds as a result of strengthened wind speeds coupled with Hg (II) inputs from atmospheric precipitation in the Intertropical Convergence Zone (ITCZ) region (Soerensen et al., 2014). While considerable research has elucidated the factors influencing the mercury sea-air exchange, few studies have comprehensively explored the repercussions of fluctuating GEM concentrations on Hg<sup>0</sup> sea-air dynamics. Given the backdrop of observed annual declines in GEM concentrations( -0.011 ± 0.006 ng m<sup>-3</sup> y<sup>-1</sup>) across most Northern Hemispheric regions from 2005 to 2020 (Feinberg et al., 2024) and particularly pronounced declines (-0.29 ng m<sup>-3</sup>) in China from 2013 to 2017 (Liu et al., 2019), conducting such study in marginal seas is essential.

Under the influence of Chinese mainland emissions, mercury pollution in its adjacent marginal seas, such as the East China Sea, Yellow Sea, and Bohai Sea, exhibited pronounced severity. The East China Sea and Yellow Sea, as semi-enclosed seas, are located in the downwind region of East Asia and serve as a major pathway for the transport of pollutants to the Pacific Ocean. The Bohai Sea, as an inland sea, has received a substantial amount of pollutants from the Chinese mainland, making it one of the most mercury-polluted seas in the world (Luo et al., 2012). By focusing on the marginal seas surrounding China, this study integrated observations from two offshore islands, one research cruise, and a coastal city, to reveal the spatiotemporal distribution characteristics of total gaseous mercury (TGM) and dissolved gaseous mercury (DGM). The impact of oceanic meteorological conditions on the atmospheric mercury over the ocean was explored, particularly examining the effects of anthropogenic sources transported from the mainland. Furthermore, we developed a method to quantify the contributions from anthropogenic sources to marine atmospheric mercury and ultimately assessed how these inputs shaped the mercury sea-air exchange dynamics.

### 2. Methods

#### 2.1 Study area

The study area, illustrated in Figure 1a, encompasses the Bohai Sea (BS), the Yellow Sea (YS), and the East China Sea (ECS). The BS, a shallow inner sea bordered by Liaoning, Hebei, and Shandong provinces, covers around 77×10<sup>3</sup> km<sup>2</sup>. The YS, situated between mainland China and the Korean Peninsula, covers around 38×10<sup>4</sup> km<sup>2</sup>. The ECS, a semi-enclosed marginal sea positioned downwind of East Asia, extends over 77×10<sup>4</sup> km<sup>2</sup>. Field measurements were conducted at Juehua Island (JHI) in the BS, approximately 10 km from Xingcheng City, Liaoning Province. Due to its proximity to the mainland, JHI experienced marked impacts from anthropogenic emissions (Li et al., 2023). Field measurements were also conducted at Huaniao Island (HNI) in the ECS, approximately 80 km from Shanghai. Although local anthropogenic emissions were negligible there, this island was frequently affected by terrestrial transport during winter and spring, when prevailing northwesterly winds dominated (Fu et al., 2018; Qin et al., 2016). A cruise campaign was conducted aboard the research vessel (R/V) Dongfanghong III. The cruise routes, as shown in Figure 1a, covered most of the YS and ECS regions. Land-based measurements were conducted at a super site

(Dianshan Lake, DSL) in the rural Shanghai Qingpu District. This super site is located at the intersection of Shanghai, Zhejiang, and Jiangsu provinces.

**Figure 1.** (a) The locations of two island sites (JHI and HNI) and one inland site (DSL) denoted by purple pentagrams. The spatial distribution of TGM concentrations over the East China Sea (ECS) and the Yellow Sea (YS) is shown along the cruise routes. The time series of TGM concentrations are measured at (b) JHI, (c) HNI, and (d) ECS+YS, respectively.

#### 2.2 TGM/GEM measurements

TGM measurements were performed utilizing a modified Tekran 2600 instrument across various locations and timeframes, i.e., JHI from December 2, 2020 to January 1, 2021, HNI from October 14 to November 4, 2020, and aboard the research vessel (R/V) Dongfanghong III from December 29, 2019 to January 16, 2020. The Tekran 2600 monitor operated similarly to Tekran 2537B, which is widely used for continuous collection and analysis of atmospheric mercury (Sprovieri et al., 2016; Landis and Keeler, 2002). During the operation of the modified Tekran 2600, atmospheric mercury was adsorbed onto the first gold trap over a 24-minute sampling period. After sampling, the mercury on the first gold trap was thermally desorbed and transferred to the second gold trap. The second trap was then analyzed by the detector during a 6-minute detection phase, resulting in an overall 30-minute sample resolution. To ensure data quality during cruise

observations, the instrument was calibrated daily using the external calibration unit Tekran 2505. Samples were pre-dried via a soda lime drying tube prior to detector entry to prevent humidity interference. Additionally, the drying tube and Teflon filter underwent replacement bi-weekly to maintain optimal performance.

GEM measurements were conducted at DSL in Shanghai from October to December, 2020, employing the atmospheric mercury monitoring system (Tekran 2537B/1130/1135) as documented in our prior study (Qin et al., 2020). Briefly, GEM was captured utilizing dual gold cartridges at a flow rate of 1.0 LPM and 5-minute intervals. Subsequently, GEM underwent thermal decomposition for detection via CVAFS. During the sampling process, rigorous quality controls were applied. Prior to sampling, denuders and quartz filters were duly prepared and cleansed adhered to Tekran technical directives. To ensure accuracy, calibration was routinely executed every 47 hours using an internal permeation source, alongside manual injections of standard saturated mercury vapor. For the Tekran 2537B, the average duplication rate between the A and B traps is 99%, with deviations between the two traps consistently below 3%. To mitigate the impact of high humidity on the instrument, samples are first passed through a soda lime drying tube for dehumidification before entering the detector. Further, the KCl-coated denuder, Teflon-coated glass inlet, and impactor plate were swapped weekly, while the quartz filters underwent monthly replacement.

It is noteworthy that TGM in the atmosphere comprises GEM and GOM (gaseous oxidized mercury). Generally, GEM constitutes over 95% of atmospheric mercury (Mao et al., 2016), particularly in the marine boundary layer, including China's marginal seas (Wang et al., 2016b; Fu et al., 2018; Ci et al., 2011; Wang et al., 2019a). Therefore, this study does not differentiate between TGM and GEM, conforming to analogous treatments in existing research (Fu et al., 2018; Ci et al., 2011).

# 2.3 DGM measurement

DGM (dissolved gaseous mercury) collection from seawater adhered to the procedure described in previous studies (Gardfeldt et al., 2003; O'driscoll et al., 2003). The sampling process involved the following steps. 1.5 L of surface seawater was collected in a Teflon bottle and subsequently transferred into a borosilicate glass bottle. An introduction of free-Hg argon at

approximately 500 ml/min purged the seawater for 60 minutes to extract the DGM onto a gold trap, aided by a soda lime tube deployed to extract water vapor prior to the gold trap. The gold trap was maintained at  $\sim$ 50°C during extraction to prevent water vapor condensation. The DGM stored in the gold trap was measured using the Tekran 2600 post-sampling. To assure quality, stringent assurance and control measures were enacted through replicated field blank experiments. DGM excised from an equivalent volume of Milli-Q water served as the analytical system blank, encompassing a total of 12 blank experiments during field samplings at JHI and HNI, as well as during the R/V measurements. The mean system blank calculated was  $2.5\pm1.3$  pg/L (n = 20), with a detection limit of 3.4 pg/L.

### 2.4 Ancillary data

At JHI, water-soluble ions in PM<sub>2.5</sub>, including sulphate (SO<sub>4</sub><sup>2-</sup>), nitrate (NO<sub>3</sub>-), ammonium (NH<sub>4</sub><sup>+</sup>), chloride (Cl<sup>-</sup>), sodium (Na<sup>+</sup>), potassium (K<sup>+</sup>), magnesium (Mg<sup>2+</sup>), calcium (Ca<sup>2+</sup>), alongside the soluble gases such as ammonia (NH<sub>3</sub>) and sulfur dioxide (SO<sub>2</sub>) were continuously monitored using an In-situ Gas and Aerosol Composition monitoring system (IGAC) (Wang et al., 2022). IGAC operated at a 1-hour temporal resolution and consisted of a wet annular denuder (WAD) and ion chromatography (IC) equipped with columns CS17 and CG17 for cations and AG11-HC and AS11-HC for anions. Ambient air was drawn into a PM<sub>2.5</sub> cyclone inlet by a built-in pump at a flow rate of 16.7 L/min. The sampled air was separated by passing through the vertically placed WAD to capture water-soluble gases, and airborne particles were collected by a steam scrubber and impact aerosol collector placed downstream. Air samples were dissolved by 30 ml ultra-pure water (18.25  $M\Omega$  cm<sup>-1</sup>) and then divided into two steams. Both aqueous samples (including particles and gases) were injected into the IC system by two separated syringe pumps for analyzing the cations and anions. For quality assurance/quality control (QA/QC) of IGAC, a standardized lithium bromide (LiBr) solution was continuously introduced into aerosol liquid samples during the campaign to validate sampling and analytical stability. Weekly calibrations were performed for the ion chromatography (IC) module using certified standard solutions, with linearity (R<sup>2</sup> > 0.99) and detection limits (LODs) validated. Black carbon (BC) in PM<sub>2.5</sub> was measured continuously using a multi-wavelength Aethalometer (AE-33, Magee Scientific, USA). Meteorological parameters were

measured using a Vaisala WXT530 surface weather station (Vaisala, Finland). Surface seawater temperature was recorded by a YSI EC300 portable conductivity meter (YSI, USA) with a resolution of 0.1°C.

At HNI, methods for analyzing meteorological parameters, BC, and surface seawater temperature mirrored those employed at JHI.

During the cruise campaign, the meteorological metrics (e.g., air temperature, wind speed/direction) and surface seawater temperature were collected from the Finnish Vaisala AWS430 shipborne weather station onboard the R/V. AE-33 was also used for BC measurements during the cruise.

At DSL, water-soluble ions in PM<sub>2.5</sub> and soluble gases were also measured by the IGAC instrument. Trace metals in PM<sub>2.5</sub> (Al, Ti, V, Cr, Mn, Fe, Co, Ni, Cu, Zn, Ga, As, Sr, Cd, Sn, Sb, Ba, Tl, Pb, and Bi) were continuously measured using an Xact multi-metals monitor (Model Xact<sup>TM</sup> 625, Cooper Environmental Services LLT, OR, USA). It operated at a flow rate of 16.7 L min<sup>-1</sup> with hourly resolution. Particles in the airflow passed through a PM2.5 cyclone inlet and were deposited onto a Teflon filter tape, then the samples were transported into a spectrometer for analysis via nondestructive energy-dispersive X-ray fluorescence.

Planetary boundary layer (PBL) height data were obtained from the Global Data Assimilation System (GDAS) archive maintained by the U.S. National Oceanic and Atmospheric Administration (NOAA), available through the READY (Real-time Environmental Applications and Display sYstem) portal (<a href="https://www.ready.noaa.gov/archives.php">https://www.ready.noaa.gov/archives.php</a>; last accessed: 11 May 2025). The dataset, featuring 1-hour temporal resolution, was processed and extracted using MATLAB R2021b (MathWorks, Natick, MA).

# 2.5 Positive matrix factorization (PMF)

The PMF model is recognized for its efficacy in elucidating sources profiles and quantifying source contributions (Paatero and Tapper, 1994). The underlying principle of PMF posits that sample concentration is dictated by source profiles with disparate contributions, mathematically represented as:

252 
$$X_{ij} = \sum_{k=1}^{P} g_{ik} f_{kj} + e_{ij}$$
 (1)

where Xij represents the concentration of the jth species in the ith sample,  $g_{ik}$  is the contribution of the kth factor in the ith sample,  $f_{kj}$  provides the information about the mass fraction of the jth species in the kth factor,  $e_{ij}$  is the residual for specific measurement, and P represents the number of factors.

The objective function, defined in Eq. (2) below, represents the sum of the squared differences between measured and modeled concentrations, weighted by concentration uncertainties. Minimizing this function allows the PMF model to determine optimal non-negative factor profiles and contributions:

260 
$$Q = \sum_{i=1}^{n} \sum_{j=1}^{m} \left( \frac{X_{ij} - \sum_{k=1}^{p} A_{ik} F_{kj}}{S_{ij}} \right)^{2}$$
 (2)

Where  $X_{ij}$  denotes the concentration of the  $j^{th}$  pollutant in the  $i^{th}$  sample,  $A_{ik}$  represents the contribution of the  $k^{th}$  factor to the  $i^{th}$  sample,  $F_{kj}$  is the mass fraction of the  $j^{th}$  pollutant in the  $j^{th}$  pollutant in  $k^{th}$  factor,  $S_{ij}$  is the uncertainty of the  $j^{th}$  pollutant in the  $i^{th}$  sample, and p is the number of factors. Detail description can be seen in the previous study (Paatero and Tapper, 1994).

TGM, air temperature (unit: Kelvin), gaseous pollutants, and major aerosol chemical species were used as inputs for the PMF model. We tested factor numbers ranging from 3 to 8, with the optimal solution determined by analyzing the slope of the Q-value versus factor count. Model stability was assessed through residual analysis, correlation coefficients between observed and predicted concentrations, and Q-value trends. A six-factor solution in DSL and a five-factor solution at JHI provided the most stable and interpretable results.

At DSL, we selected observational data from October to December, 2020 (totaling 1,080 valid data points) for PMF modeling to align with the HNI observational campaign. At JHI, observational data from December 2 to 30, 2020 (totaling 675 valid data points) were used for PMF analysis.

# 2.6 Sea-air exchange flux

The sea-air exchange fluxes of Hg<sup>0</sup> were calculated via the following equation (Andersson et al., 2008a; Wanninkhof and Oceans, 1992; Wangberg et al., 2001):

278 
$$F = K_W(C_W - C_a/H')$$
 (3)

where F is the sea-air exchange flux, Kw represents the gas exchange velocity, Cw and Ca represent

the DGM concentration in seawater and the TGM concentration in the atmosphere, respectively, H'
is the dimensionless Henry's law coefficient of Hg<sup>0</sup> between the atmosphere and seawater. K<sub>w</sub> is
calculated as follows (Soerensen et al., 2010b; Nightingale et al., 2000).

283 
$$K_{\rm w} = 0.31 u_{10}^2 (S_{C_{\rm H}\sigma}/660)^{-0.5} \tag{4}$$

where  $u_{10}$  is 10-meter wind speed,  $S_{C_{Hg}}$  is the Schmidt number of  $Hg^0$ , 660 is the Schmidt number of  $CO_2$  in 20 °C seawater (Poissant et al., 2000). The Schmidt number for Hg ( $S_{CHg}$ ) was calculated

as:

$$S_{C_{Hg}} = v/D_{Hg} \qquad (5)$$

where v is seawater kinematic viscosity (Wanninkhof, 2014) and D<sub>Hg</sub> is the diffusion coefficient of

- Hg (Kuss et al., 2009).
- H' is calculated as follows (Andersson et al., 2008a).

$$H' = \exp(-2403.3/T + 6.92)$$
 (6)

Where T is the surface seawater temperature in K.

# 3. Results and Discussions

# 3.1 Characteristics of TGM over Chinese marginal seas

Figure 1b-d shows the time series of TGM concentrations measured during three field campaigns, including December 2, 2020 to January 1, 2021 at Juehua Island (JHI), October 14, 2020 to November 4, 2020 at Huaniao Island (HNI), and December 29, 2019 to January 16, 2020 over the Yellow Sea and East China Sea (YS/ECS). The mean TGM concentrations during the three periods were  $2.32 \pm 1.02$  ng/m³,  $1.85 \pm 0.74$  ng/m³, and  $2.25 \pm 0.66$  ng/m³, respectively. TGM at JHI exhibited pronounced fluctuations, frequently surpassing high values of 6 ng/m³, which was attributed to the enhanced coal combustion for residential heating in winter (Li et al., 2023). Conversely, TGM at HNI and across the YS/ECS demonstrated less fluctuations, with concentrations predominantly remaining below 6 ng/m³. The cruise campaign unveiled the spatial distribution of TGM over the ocean (Figure 1a), generally showing its decreasing trend with the increased distance away from the continent. Specifically, hot spots were observed in the eastern oceanic region of Jiangsu province, the Changjiang estuary, and the outer sea close to the Hangzhou

Bay. The continental outflows likely explained this phenomenon. The mean TGM concentrations reached  $2.36 \pm 0.65$  ng/m³ and  $2.16 \pm 0.81$  ng/m³ over the Yellow Sea and East China Sea, respectively, significantly higher than the background level in the Northern Hemisphere  $(1.58 \pm 0.31 \text{ ng/m}^3)$  (Bencardino et al., 2024) and also surpassing measurements recorded in the other open ocean areas such as the South China Sea  $(1.52 \pm 0.32 \text{ ng/m}^3)$ , Mediterranean Sea  $(1.8 \pm 1.0 \text{ ng/m}^3)$ , Bering Sea  $(1.1 \pm 0.3 \text{ ng/m}^3)$ , Pacific Ocean  $(1.15 - 1.32 \text{ng/m}^3)$ , and Atlantic Ocean  $(1.63 \pm 0.08 \text{ ng/m}^3)$  (Laurier and Mason, 2007; Soerensen et al., 2010a; Mastromonaco et al., 2017; Kalinchuk et al., 2018; Wang et al., 2019b). 72h air mass backward trajectory analyses revealed that air masses over the YS predominantly originated from Liaoning and Inner Mongolia province in northern China, whereas trajectories over the ECS were largely dispersed across the ocean and Eastern China (Figure S1). This divergence may be one of the reasons why the TGM concentration in the YS was higher than that in the ECS.

The diurnal variations of TGM concentrations along with ambient temperature and sun flux during the three periods are displayed in Figure 2. At HNI, TGM commenced increasing at 7:00 a.m., peaking at around 2.44 ng/m<sup>3</sup> by 12:00, subsequently declining and stabilizing post 6:00 p.m. The mean TGM concentration during daytime (06:00-18:00) (2.00  $\pm$  0.80 ng/m<sup>3</sup>) surpassed that of nighttime (1.66  $\pm$  0.40 ng/m<sup>3</sup>) (t test, p < 0.001). The TGM diurnal pattern displayed strong concordance with temperature and solar flux (Figure 2a), indicative of significant impacts from natural sources (Osterwalder et al., 2021; Huang and Zhang, 2021; Mason et al., 2001). At JHI (Figure 2b), TGM also rose around early morning and peaked at 2.65 ng/m<sup>3</sup> by 10:00 a.m., with nocturnal levels markedly increasing from 2.12 ng/m<sup>3</sup> at 6:00 p.m. to 2.60 ng/m<sup>3</sup> at 11:00 p.m. During daytime, TGM generally showed consistent variation with temperature and sun flux, indicating the influence of natural mercury release. However, the notable frequency of nocturnal peaks suggested that in addition to natural sources, TGM measured at JHI was also significantly affected by anthropogenic sources and unfavorable atmospheric diffusion conditions, specifically from coal combustion for the winter residential heating in northern China (Li et al., 2023). The diurnal pattern of TGM throughout the cruise campaign diverged from those of HNI and JHI, lacking a consistent relationship with temperature and sun flux. This was mainly due to that the cruise sampling was variable in both the temporal and spatial scale.

**Figure 2.** Diurnal variations of TGM, ambient temperature and sun flux at (a) HNI, (b) JHI, and (c) the YS/ECS cruise, respectively.

Positive correlations between TGM concentrations and ambient temperature at both HNI and JHI were observed, yielding R<sup>2</sup> values of 0.53 and 0.60, respectively (Figure 3a&3b). Since temperature played a crucial role in Hg<sup>0</sup> release from natural surfaces (Lindberg et al., 1998; Poissant et al., 2000), the evident correlation between TGM and temperature exemplified the significant effects of natural surface emissions.

Positive correlations were also observed between TGM, relative humidity, and wind speed at both HNI and JHI (Figure 3). The positive correlation between humidity and TGM may be due to the fact that high humidity is typically associated with the stable atmospheric stratification, which promoted the accumulation of TGM. As for wind speed, it is a key parameter influencing air-sea exchange in the double-membrane theory model (Wanninkhof, 1992). For example, Soerensen et al. (2014) found a 2–4 times greater Hg<sup>0</sup> flux due to the high wind speed in the Intertropical Convergence Zone (ITCZ) region.

At HNI, TGM increased concurrently with rising Planetary Boundary Layer (PBL) heights

from around 380 to 1000 m, yet decreased with further increase in PBL beyond around 1000 m (Figure 3a). This observed diurnal pattern of TGM may primarily stem from the interplay between temperature-driven natural surface emissions and atmospheric dilution effects. When the PBL height was below 1 km, its increase coincided with rising temperature. Under these conditions, the enhancement of natural surface emissions due to temperature outweighed the dilution effect caused by the developed PBL, leading to increased TGM concentrations. Afterwards, as the PBL height continued to rise, the dilution effect gradually surpassed the temperature-driven emission enhancement, resulting in a decline of TGM concentrations. In contrast, the similar phenomenon lacked manifestation at JHI, where TGM concentrations decreased with the increase of PBL (Figure 3b). Due to the significantly lower marine mercury emissions in the BS (Wang et al., 2020) than in the ECS (Wang et al., 2016a), this phenomenon was likely ascribed to that the natural release around JHI was weaker than that around HNI, thus the dilution effect of elevated PBL overwhelmed the effect of natural surface emissions. Compared to HNI and JHI, the cruise campaign showed almost no relationship between TGM and temperature, relative humidity, wind speed, or PBL height were identified (Figure 3c), which shared similar reasons as discussed in the diurnal variation of TGM.

**Figure 3.** Relationship between TGM concentration and temperature, relative humidity, planetary boundary layer height, and wind speed at (a) HNI, (b) JHI, and (c)YS/ECS, respectively.

### 3.2 Influence of continental outflows on marine TGM

The potential source regions of TGM at HNI and JHI are illustrated in Figure S2. At HNI, TGM mainly derived from the lands of Jiangsu Province and vast coastal waters of the East China Sea. While at JHI, the hot spots of TGM were mainly located in the southern Mongolia and Beijing-Tianjin-Hebei regions. This indicated that the relatively high TGM concentrations at the coastal islands were closely related to the continental outflows. Using HNI as an example, Figure S3 compares the daily mean TGM concentration at HNI with the daily mean concentrations of CO, SO<sub>2</sub>, and PM<sub>2.5</sub> in nearby coastal cities including Zhoushan, Ningbo, Jiaxing, Shanghai, and Ningbo. Consistently temporal variations were observed between TGM and these pollutants, particularly for

the peak concentrations, further confirming that offshore TGM concentrations were significantly influenced by continental outflows.

To assess the impact of anthropogenic sources on marine TGM, the daily TGM concentrations at HNI and DSL (a suburban site in the Yangtze River Delta, Figure 1a) were compared (Figure 4a). Their concentration time series exhibited moderate agreement, suggesting potential land-sea interactions. Furthermore, the correlation between TGM and BC at DSL was pronounced (R<sup>2</sup>=0.56, Figure 4b). This was expected, as BC primarily originated from fossil fuels combustion (Li et al., 2021; Briggs and Long, 2016), which was also the major anthropogenic source of TGM (Pacyna et al., 2006; Streets et al., 2011; Liu et al., 2019). In contrast, the correlation between TGM and BC at HNI was much weaker (R<sup>2</sup>=0.34, Figure 4c). Being an offshore site, HNI could be more strongly influenced by natural sources than DSL.

To evaluate the relative importance of anthropogenic and natural sources to TGM, the ratio of TGM/BC was introduced as a qualitative index. Since TGM and BC shared common anthropogenic sources, and TGM had additional natural sources, an increase in the TGM/BC ratio may indicate the growing importance of natural source contributions, and vice versa. Figure 4d shows that the TGM/BC ratio at DSL (mean of 1.6 ng µg<sup>-1</sup>) was substantially lower than that at HNI (5.2 ng µg<sup>-1</sup>). On one hand, lower contribution of anthropogenic sources to TGM in the coastal environment compared to the urban environment was expected. On the other hand, BC deposited more quickly than TGM, thus also elevating the TGM/BC ratios at locations far from emission sources. The cruise measurement illustrated the spatial distribution of the TGM/BC ratio over YS/ECS (Figure 4e). In the East China Sea, the TGM/BC ratio increased with increasing distances away from the coasts. For instance, the TGM/BC ratio near the coasts ranged from 0.3 to 5.2 ng μg<sup>-1</sup>, while offshore values generally fluctuated between 8.6 and 22.9 ng µg<sup>-1</sup>. This indicated the contribution of natural sources to TGM obviously increased over the open ocean waters. However, this spatial trend was not observed in the Yellow Sea. As depicted in Figure 4e, the very northern, western, and eastern cruise legs in the Yellow Sea showed relatively low TGM/BC ratios compared to the other cruise periods. This phenomenon should be due to the Yellow Sea being a comparatively enclosed basin, as these cruise legs above were geographically close to Liaoning province in northeast China, the North China Plain, and the Korean Peninsula. Thus, more influences from the terrestrial emissions induced

the low TGM/BC ratios.

**Figure 4.** (a) Comparison of the daily TGM concentrations between DSL and HNI; Correlation between TGM and BC at (b) DSL and (c) HNI; (d) Comparison of the TGM/BC ratio between DSL and HNI; (e) Spatial distribution of the TGM/BC ratio along the cruise routes over the ECS and YS.

### 3.3 Quantification of anthropogenic vs. marine sources to TGM

Based on the discussions above, it is essential to disentangle the anthropogenic and natural sources of atmospheric mercury. Here, the PMF model was employed for the comprehensive dataset obtained at DSL and JHI, respectively. Considering the direct correlation between temperature and natural release of atmospheric mercury (Wang et al., 2014; Zhu et al., 2016) and the indirect correlation between ammonia and natural release of atmospheric mercury (Qin et al., 2019), we utilized temperature and ammonia as indicators of natural atmospheric mercury sources, which has been proven feasible (Qin et al., 2020). Inputs for PMF also encompassed SO<sub>4</sub><sup>2-</sup>, Cl<sup>-</sup>, NO<sub>3</sub><sup>-</sup>, Na<sup>+</sup>, NH<sub>4</sub><sup>+</sup>, K<sup>+</sup>, Mg<sup>2+</sup>, Ca<sup>2+</sup>, SO<sub>2</sub>, NO, Pb, Fe, K, Cr, Se, Ca, V, Mn, As, and Ni. While running the PMF model, we tested the number of factors from three to eight and determined the optimal solutions through analyzing the slope of the Q values in relation to the number of factors. The analyses revealed that a six-factor solution for DSL and a five-factor solution for JHI produced the most robust and coherent interpretations.

**Figure 5.** Source apportionment of TGM at (a) DSL and (b) JHI; Linear relationship between BC and anthropogenic TGM at (c) DSL and (d) JHI.

As detailed in Figure 5a, six distinct factors were resolved by PMF at DSL. The factor characterized by high loadings of temperature, NH<sub>3</sub>, and TGM represented the natural surface emissions of mercury. A second factor with notable loading of V and moderate loading of Ni was ascribed to be shipping emissions, as V has been considered a typical tracer of heavy-oil combustion, which is commonly used in marine vessels (Viana et al., 2009). The dust and cement production was associated with a factor exhibiting prominent Ca loading. The factor displaying high loading of Cr and moderate loadings of Fe and Mn was attributed to iron and steel production. Another factor, categorized by elevated NO levels, was linked to vehicle emissions. Finally, the factor with high loadings of SO<sub>4</sub><sup>2-</sup>. Pb, K, Se, and As was indicative of coal combustion. PMF results indicated that the contributions of anthropogenic and natural sources to TGM were approximate 48% and 52% at DSL, respectively. By applying the same PMF modeling strategy at JHI, the contributions of anthropogenic and natural sources to TGM at JHI were 59% and 41%, respectively (Figure 5b). The source apportionment results signified substantial influences of both human and natural factors on TGM levels, with their contributions being nearly equivalent. Furthermore, correlation analysis was

conducted between the absolute contribution of anthropogenic sources to GEM and BC, yielding strong correlations at both DSL (Anthropogenic TGM =  $(0.424 \pm 0.014)*BC + (0.171\pm 0.032)$ ,  $R^2 = 0.88$ , Figure 5c) and JHI (Anthropogenic TGM =  $0.362 \pm 0.009)*BC + (0.431 \pm 0.044)$ ,  $R^2 = 0.86$ , Figure 5d), respectively. It should be noted that BC was not included in PMF modeling, thus the robust relationship between anthropogenic GEM and BC suggested that BC can serve as a viable indicator for quantifying anthropogenic contributions to TGM.

To validate the robustness of this relationship in different years, we derived the relationship between anthropogenic GEM and BC at DSL before 2020 based on the same methodology. It can be found that the regression equation during the winter of previous years was close to that obtained during this study period (Figure S5). In fact, the mercury emissions (Feng et al., 2024) and black carbon emissions (Geng et al., 2024) were quite stable in the neighboring years of this study period. For instance, China's anthropogenic GEM emissions in 2019 and 2020 were 194.2 tonnes and 191.8 tonnes, respectively, showing negligible changes. Thus, it can be assumed that the relationship between anthropogenic GEM and BC remained relatively constant.

Based on the results above, the regression formulas obtained from DSL and JHI were further applied to the cruise observation for the purpose of differentiating the anthropogenic and natural fractions of TGM over the ocean. The following criteria were applied. If the air mass backward trajectories (purple segments in Figure S5) primarily originated from northern China, the JHI-derived formula was employed; If the air mass backward trajectories (green segments in Figure S5) passed through the Yangtze River Delta region or hovered over the East China Sea, the DSL-derived formula was enacted. Due to the uncertainties of regression slopes and intercepts of the regression formulas, this approach caused around 5% uncertainties on differentiating the anthropogenic and natural fractions of TGM.

Time-series of mass concentrations of anthropogenic and natural TGM in different coastal and oceanic regions after applying the above equations are shown in Figure S6. Concentrations of anthropogenic TGM at HNI, JHI, ECS, and YS were  $0.61 \pm 0.29$  ng/m<sup>3</sup>,  $1.28 \pm 0.75$  ng/m<sup>3</sup>,  $0.59 \pm 0.41$  ng/m<sup>3</sup>, and  $0.92 \pm 0.25$  ng/m<sup>3</sup>, respectively. And the concentrations of natural TGM were  $1.19 \pm 0.45$  ng/m<sup>3</sup>,  $0.88 \pm 0.26$  ng/m<sup>3</sup>,  $1.57 \pm 0.53$  ng/m<sup>3</sup>, and  $1.38 \pm 0.51$  ng/m<sup>3</sup> at the four locations above, respectively. To ensure the reliability of these results, the relationship between natural TGM

and temperature was explored, yielding R<sup>2</sup> of 0.92, 0.76, 0.75, and 0.63 at HNI, JHI, ECS, and YS, respectively (Figure 6). The correlation was significantly stronger than that between the total TGM and temperature, particularly in the ECS and YS regions, where no correlations were observed (Figure 3c). This proved that the quantitative method established above was reliable.

**Figure 6.** (a-d) Relationship between natural TGM and temperature at HNI, JHI, ECS, and YS, respectively. (e) Spatial distribution of the relative contributions of anthropogenic and natural sources to TGM concentrations along the cruise routes. The mean contributions over JHI, HNI, DSL, ECS, and YS are denoted by the pie charts.

The anthropogenic contributions to TGM along the cruise routes are plotted in Figure 6e, demonstrating significantly higher values near the coastal zones compared to the open ocean areas. In details, anthropogenic contributions to TGM near the East China Sea coastal zones reached as high as 60-88%, while the contributions diminished quickly to 15-25% over the open oceans. From the northern oceanic regions to the southern counterparts, the contribution of anthropogenic sources to TGM generally exhibited a decreasing trend, with values of around 59%, 40%, 34%, and 27% over JHI, YS, HNI, and ECS, respectively. In comparison, previous isotope-based source apportionment studies have revealed anthropogenic contributions of 29% and 42% to TGM in

remote areas like Changbai Mountain and Ailao Mountain (Wu et al., 2023). In general, the isotopebased results indicated that the relative contributions of anthropogenic emissions to surface GEM in remote China and urban China were around 30% and 49%, respectively (Fu et al., 2021; Feng et al., 2022; Wu et al., 2023). Notably, the anthropogenic contributions to TGM in the Yellow Sea, East China Sea, and Huaniao Island from this study aligned closely with isotope-derived values from China's remote regions, while the DSL findings corresponded with urban isotope results. The elevated contribution observed at JHI (59%) may be attributed to its proximity to the mainland (only 10 km away) and the sampling period occurring during the winter heating season, where continental transport influences were significant (Li et al., 2023). Furthermore, the values obtained in this study fell within comparable ranges to modeling study estimates (typically 33% to 41% on average) (Chen et al., 2014; Wang et al., 2018). As shown in Figure S1, the backward trajectories over the Yellow Sea segment were primarily influenced by air masses from the North China Plain and Liaoning Province. The relatively higher contribution of anthropogenic sources to the Yellow Sea during the cruise was likely attributable to the continental transport from northern China. In addition, during this cruise, the seawater temperature of the Yellow Sea was significantly lower than that of the East China Sea, which was unfavorable for the natural release of mercury.

# 3.4 Characteristics of sea-air exchange of mercury in various oceans

To determine the sea-air exchange flux of mercury, DGM (dissolved gaseous mercury) concentrations in seawater were measured at all sampling sites during the cruise (Figure S7). Figure S7 delineates the time-series of DGM observed at JHI, YS, HNI, and ECS, with mean concentrations of  $21.3 \pm 4.8$ ,  $29.9 \pm 6.1$ ,  $42.0 \pm 9.4$ , and  $39.7 \pm 10.9$  pg/L, respectively. The DGM concentrations measured during this winter cruise campaign (22.9-39.7 pg/L) were significantly lower than those recorded previously during summer and fall in similar regions (52.4-63.9 pg/L) (Ci et al., 2011; Ci et al., 2015; Wang et al., 2016a), indicating a noticeable seasonal variation in DGM concentrations in the ECS and YS. This seasonal variation pattern of seawater DGM, with lower levels in winter compared to summer and autumn, can be attributed to the dynamic equilibrium between competing redox processes. This equilibrium can be represented as:  $Hg^{2+}$  + photoreductants  $\rightleftharpoons$  DGM + photo-oxidants (O'driscoll et al., 2006). During warmer seasons, higher

temperature accelerated the volatilization of DGM from seawater, and also drove the equilibrium toward Hg<sup>2+</sup> reduction to replenish the lost DGM. Therefore, DGM concentrations in seawater were usually lower in winter due to suppressed redox processes. Spatially, DGM concentrations in the ECS were higher than those in the YS, likely due to the significantly higher sea surface temperature in the ECS (mean: 14.8 °C) compared to the YS (mean: 4.1 °C) during the cruise campaign. Higher temperature not only favored the production of DGM in seawater (Costa and Liss, 1999; Andersson et al., 2011; Mason et al., 2001) but also promoted the escape of DGM from the water surface into the atmosphere (Osterwalder et al., 2021; Huang and Zhang, 2021).. Additionally, we observed that DGM concentrations were higher in coastal waters, particularly near the Yangtze River Estuary, where the concentration reached 51.4 pg/m³. This suggested that continental inputs, such as river discharge, had a significant influence on DGM levels in nearshore waters (Chen et al., 2020; Kuss et al., 2018; Liu et al., 2016).

**Figure 7.** (a-c) Sea-air exchange fluxes of mercury at JHI, YS, and ECS, respectively. (d) The spatial distribution of sea-air exchange flux of Hg<sup>0</sup> during the cruise.

Figure 7 shows the time-series and spatial distribution of sea-air exchange fluxes of mercury within the BS (represented by JHI), YS, and ECS, which were  $0.17\pm0.38$ ,  $1.10\pm1.34$ , and  $3.44\pm3.24$  ng m<sup>-2</sup> h<sup>-1</sup>, respectively (Table S1). It was evident that the Hg<sup>0</sup> fluxes during winter in the ECS was the highest, followed by the YS and the BS. This finding coincided with the discussions above that natural TGM exhibited much higher concentrations in the ECS and YS than in the BS (Section 3.3). BS acted as a weak mercury source region and even a mercury sink sometimes (negative flux in Figure 7a). Due to the higher concentrations and contributions of anthropogenic TGM in the BS, the release of mercury from the ocean was significantly suppressed, which likely explained the relatively low sea-air exchange flux of mercury there.

Overall, mercury fluxes observed during winter were lower than previous studies in other seasons, e.g., 4.6±3.6 ng m<sup>-2</sup> h<sup>-1</sup> in ECS during summer (Wang et al., 2016a), 3.07±3.03 ng m<sup>-2</sup> h<sup>-1</sup> in YS during spring (Wang et al., 2020), and 0.59±1.13 ng m<sup>-2</sup> h<sup>-1</sup> in BS during fall (Wang et al., 2020).

# 4. Conclusions and implications

This study elucidated the effects of anthropogenic sources on atmospheric mercury concentrations across various marginal seas and the subsequent influence on sea-air exchange dynamics of mercury. Through comprehensive observations across island, cruise, and terrestrial settings, we delineated atmospheric mercury distribution characteristics within the Chinese marginal seas. The relationships between TGM and various environmental parameters suggested the significance of natural sources in constraining oceanic atmospheric mercury levels. Notably, TGM peaks recorded at terrestrial and island sites exemplified the influence of continental outflows on the marine TGM. The introduction of the TGM/BC ratio functioned as a qualitative proxy for assessing the extent of anthropogenic contributions. Furthermore, we articulated a quantitative methodology for assessing anthropogenic contributions to marine atmospheric mercury, revealing that these sources contributed 59%, 40%, and 27% to atmospheric mercury levels across the Bohai

Sea, Yellow Sea, and East China Sea, respectively. The winter sea-air exchange fluxes of mercury in these three seas were estimated as 0.169, 1.100, and 3.442 ng m<sup>-2</sup> h<sup>-1</sup>, respectively, In regions where anthropogenic emissions were intense, sea-air exchange fluxes of mercury were evidently suppressed.

Conducting atmospheric mercury measurements over oceans presented considerable complexities compared to terrestrial observations, further compounded by challenges associated with determining atmospheric mercury sources in the oceanic environment. This study established a quantitative method grounded in extensive observations encompassing terrestrial, island, and marine contexts, facilitating estimations of anthropogenic contributions to atmospheric mercury solely predicated on atmospheric mercury and black carbon data. This methodology may offer valuable insights for analogous analyses of atmospheric mercury and other pollutants across diverse oceanic regions globally. These insights contribute to a deeper understanding of the biogeochemical cycle of mercury and enhance our ability to evaluate its impacts on marine ecosystems and human health.

#### **Data Availability Statement**

The raw data generated in this study have been uploaded to Zenodo

(https://doi.org/10.5281/zenodo.14847622).

# **Author contributions**

KH designed this study. XQ, HL, JC, XW, GW, CL, DL, JH, and YL performed data collection.

XQ and KH performed data analysis and wrote the paper. All have commented on and reviewed the

paper.

# **Competing interests**

The authors declare that they have no conflict of interest.

# Acknowledgments

This work was financially supported by the National Key R&D Plan Programs

- (2023YFE0102500, 2018YFC0213105), the National Science Foundation of China (42175119,
- 42361144711), and the Joint Research Fund of the "Island Atmosphere and Ecology" Category IV
- Peak Discipline (ZD202502). Junjie Wei, Hao Ding, and Qiongzhen Wang acknowledge financial
- support from Central Guiding Local Science and Technology Development Fund Projects
- (No.2023ZY1024).

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
