# Peer review of "Quantification of anthropogenic and marine sources to atmospheric"

_EGUsphere, 2025_

## Author Comment (AC1)

**Response to Reviewer #1's Comments**

Anonymous Referee #1:

Major comments:

The authors conducted about one month (12/2/2020-1/1/2021), 3 weeks (10/14/2020-11/4/2020), and 2 weeks (12/29/2019-1/16/2020) of TGM measurements at two island sites, JHI and HNI, and on a cruise ship. They estimated anthropogenic contributions to ambient concentrations of TGM using PMF and linear regression analysis. They also estimated sea-air exchange flux of Hg° using an air-water exchange flux equation (Wanninkhof, 1992, JGR). Over the past couple of decades since the Tekran series has been deployed globally, numerous long-term datasets of speciated, operationally defined, mercury concentrations have been available and used to study atmospheric Hg cycling, which has generated a large body of research in the literature. While the authors performed a comprehensive analysis with what they got, the short-term nature of their datasets limited their ability to provide substantial insights into atmospheric Hg budgets. The study also presents several methodological concerns.

We sincerely thank for your in-depth comments and helpful suggestions on this manuscript. Based on the specific comments, we have responded to all the comments point-by-point and made corresponding changes in the manuscript as highlighted in the track change mode. You have raised a number of issues and we quite agree. We feel the substantial revisions based on your comments have greatly improved the quality of this manuscript. Please check the detailed responses to all the comments as below.

Specific comments:

1. PMF application and interpretation issues. There are multiple concerns regarding the PMF application and interpretation:

1.1 The authors used PMF to identify the factors contributing to ambient TGM at DSL and JHI. DSL was a long-term (2015-2019) monitoring site over land near the coast in Shanghai, while the field campaign at HNI, an island site near DSL, took place a year later. The authors should clearly specify the time periods used in their PMF analysis for the two sites in Section 2.5.

Response: Thanks for your comments regarding the PMF modeling. We do agree that inconsistent observation periods can affect the results. In this regard, we now use observational data which are consistent between DSL and JHI. We have clearly specified the time periods used in the PMF analysis for the two sites in Section 2.5.

The revised manuscript includes the following additions in Line 289-292.
At DSL, we selected observational data from October to December, 2020 (totaling 1,080 valid data points) for PMF modeling to align with the HNI observational campaign. At JHI, observational data from December 2 to 30, 2020 (totaling 675 valid data points) were used for

1.2 The study assumes that the empirical relationship between anthropogenic TGM contributions and BC concentrations (derived from 2015–2019 land-based data) is applicable to TGM data at an island site over 100 km away and one year later. This assumption is highly questionable, as empirical relationships may not necessarily hold across different locations and timeframes. At the very least, the authors should acknowledge the potential uncertainty introduced by this approach.

Response: Thanks for your comments. We do agree with you that inconsistent observational periods may result in great uncertainties. As responded to the previous comment, we have now used consistent observational periods at both sites to avoid this issue. As for the potential uncertainty introduced by this approach, we now consider the uncertainty of the derived regression equations, which is described in the next response.

1.3 The size of the dataset for JHI appears too small for PMF. It was a month-long campaign. What was the temporal resolution they had for their datasets? Fig. 5d seems to show about 20 data points. If this represented the number of data points for a single variable used for PMF analysis, then their results would be questionable. The authors should reference Zhang et al. (2009, https://doi.org/10.1016/j.atmosenv.2009.07.009), which discusses the minimum sample size required for PMF applications. That reference is just an example from a large body of literature on the topic. In fact, I am wondering, if the authors used the 2015-2019 data for DSL PMF analysis, how come only a handful of data points were shown in Fig. 5c?

Response: We apologize for the lack of clarity in the original manuscript. Regarding the original Figure 5c and 5d, the data points represent daily averages, that's why only a small number of points is shown in the figure.

For Juehua Island (JHI), the data used for PMF analysis were measured from December 2 to 30, 2020, with a 1-hour temporal resolution, totaling 675 data points. For Dianshan Lake (DSL), the PMF analysis utilized observational data from October to December, 2020, with a 2-hour temporal resolution, totaling 1080 data points. Thus, the dataset sizes for both sites meet the requirements for PMF analysis.

In the revised Figure 5c and 5d as shown below, we have now used the original dataset for plotting. It is found that the regression equations based on the hourly datasets are quite close to those based on the daily datasets. In addition, the revised Figure 5c and 5d also show the regression equations with uncertainties. In the revised manuscript, we have re-calculated the contributions of anthropogenic vs. natural TGM and added the uncertainties based on this approach.

[Figure]

In Line 501 - 504, the concentrations of anthropogenic and natural TGM are revised based on the newly derived formulas.

In Line 497-499, the following writings are added.
Due to the uncertainties of regression slopes and intercepts of the regression formulas, this approach caused around 5% uncertainties on differentiating the anthropogenic and natural fractions of TGM.

1.4 The rationale behind selecting certain tracers is unclear. Why did the authors use V for the shipping emission tracer? The tracers for the cement industry were Ca and Fe, which could very well be indicative of dust. Why was BC not used in the PMF analysis?

Response: Thanks for your comments. V was chosen as an indicator for shipping emissions because V has been recognized as a typical tracer of heavy-oil combustion, which is commonly used in marine vessels (Viana et al., 2009). As such, previous studies have widely adopted V as a key marker for ship-related emissions (Chang et al., 2018; Zhao et al., 2013; Pandolfi et al., 2011).

In the revised manuscript, we have added the following clarification in Line 474-475.
"V has been considered a typical tracer of heavy-oil combustion, which is commonly used in marine vessels (Viana et al., 2009)."

Regarding calcium (Ca), it was a major airborne component from cement production since there are a lot of cement industries in China. However, it is indeed true that Ca and Fe are well indicative of dust. In the revision, we have labeled this factor as "dust and cement production". (Line 475).

As for Black carbon (BC), if it was also included in the PMF analysis, it is expected that BC always correlated with anthropogenic TGM since BC and anthropogenic TGM would be assigned to the same factor. In this study, we intentionally excluded BC from the PMF analysis and

retained it as an independent variable, thus the PMF results and BC measurements were independent with each other. Then, we found out that anthropogenic TGM derived from PMF correlated strongly with BC at two sites. On one hand, this corroborated the reasonability of the PMF modeling results. On the other hand, the results suggested the relationship between anthropogenic TGM and BC was robust when evaluated at two sites.

2. TGM/BC ratio. The authors highlighted the TGM/BC ratio in the abstract as a key finding. However, this ratio appears to be just another variable rather than a novel result that provides additional insights.

Response: Thanks for your suggestion. In the revised manuscript, we have removed the description of the TGM/BC ratio from the abstract.

3. Sea-Air exchange flux calculations. The study recalculates sea-air exchange flux after removing anthropogenic contributions from ambient data. However, the purpose of this recalculation remains unclear. This issue also relates to the statement in the abstract (Lines 39–40), which needs further clarification.

Response: Thanks for your comments. In this study, we recalculated the air-sea exchange flux after removing anthropogenic contributions to quantify the impact of human-driven sources on oceanic mercury release. The purpose of this recalculation is to assess potential changes in the marine mercury flux under the scenario of reduced anthropogenic emissions.

In the revision, this sentence (Line 44-47) is revised as "To assess the potential impact of anthropogenic emissions on the sea-air exchange fluxes of mercury, anthropogenic contributions to TGM were artificially removed, then the fluxes would be increased by 207.1% in the Bohai Sea, 33.4% in the Yellow Sea, and 6.5% in the East China Sea, respectively.".

4. Unsupported assertions. Assertions throughout the manuscript lack supporting evidence or citations. Below are a few examples:

4.1 Line 256: The term "elucidation" is misleading, as the statement is purely speculative.

Response: We agree with the reviewer that the original phrasing overstated the conclusiveness of the analysis. The sentence (Line 341-342) has been revised as "This divergence may be one of the reasons why the TGM concentration in the YS was higher than that in the ECS."

4.2 Lines 261–263, 265–266 (JHI), 266–268, 287: Assertions require supporting evidence or references.

(1) Lines 261–263

Response: From the calculation of oceanic mercury release fluxes, under otherwise constant conditions, higher temperatures lead to greater mercury release from the ocean (Wanninkhof and Oceans, 1992). This is because $Hg^0$ inherently possesses high volatility. Elevated temperatures intensify molecular thermal motion, accelerating the volatilization of mercury from the liquid phase to the gas phase.

From the perspective of $Hg^0$ production mechanisms in seawater, the production of $Hg^0$ in seawater primarily originates from the photochemical reduction of divalent mercury ($Hg^{2+}$)

(Costa and Liss, 1999; Andersson et al., 2011), which is closely linked to parameters such as light intensity and temperature (Ci et al., 2016; Mason et al., 2001).

Therefore, in the revised manuscript, we have cited relevant references as supporting evidence, as demonstrated in Line 348-350.

"The TGM diurnal pattern displayed strong concordance with temperature and solar flux (Figure 2a), indicative of significant impacts from natural sources (Osterwalder et al., 2021; Huang and Zhang, 2021; Mason et al., 2001)."

(2) Lines 265–266 (JHI)

Response: The answer to this question aligns with that of the previous response.

(3) Lines 266–268

Response: We have cited field measurements from the HNI (Fu et al., 2018) to support anthropogenic influences on coastal TGM. Also, field measurements from JHI (Li et al., 2023) are cited to support the are primarily influenced by coal combustion for heating in winter.

(4) Line 287

We recognized that the conclusion in the original text—"Thus, relative humidity and wind speed were verified as critical factors influencing the levels of TGM as similar as temperature"—was overly assertive. Therefore, in the revised version (Line 378-379), we have rephrased this statement as "This may explain the observed positive correlations between humidity, wind speed, and TGM concentrations."

4.3 Lines 281–300: This paragraph is speculative and lacks supporting evidence.

Response: Regarding the conclusion in lines 281–282, "the evident correlation between TGM and temperature exemplified the significant effects of natural surface emissions," we have already addressed this in the response of question **4.2(1)** and supplemented it with supporting literature.

The statement in lines 287–288, "Thus, relative humidity and wind speed were verified as critical factors influencing the levels of TGM as similar as temperature," has been revised in the answer of **question 4.2(4)** to be "This may explain the observed positive correlations between humidity, wind speed, and TGM concentrations."

For the unclear explanation in lines 291–294 regarding the phenomenon where TGM concentrations initially rise and then decline with increasing PBL (Planetary Boundary Layer) height, we have refined the reasoning in the revised version (Line 384-390):
"This observed diurnal pattern of TGM may primarily stem from the interplay between temperature-driven natural surface emissions and atmospheric dilution effects. When the PBL height was below 1 km, its increase coincided with rising temperature. Under these conditions, the enhancement of natural surface emissions due to temperature outweighed the dilution effect caused by the developed PBL, leading to increased TGM concentrations. Afterwards, as the PBL height continued to rise, the dilution effect gradually surpassed the temperature-driven emission enhancement, resulting in a decline of TGM concentrations."

Finally, to support the conclusion in lines 295–297—"This was likely ascribed to that the natural release around JHI was weaker than that around HNI, thus the dilution effect of elevated PBL overwhelmed the effect of natural surface emissions"—we have incorporated observational data

from the BS (Wang et al., 2020) and ECS (Wang et al., 2016a). These data corroborate that marine mercury emissions in the BS are significantly lower than those in the ECS.

Line 395-396: Due to the significantly lower marine mercury emissions in the BS (Wang et al., 2020) than in the ECS (Wang et al., 2016a), this phenomenon was likely ascribed to that the natural release around JHI was weaker than that around HNI, thus the dilution effect of elevated PBL overwhelmed the effect of natural surface emissions.

4.4 Line 319: The claim appears overstated.

Response: We appreciate the reviewer's critical perspective. The original phrasing has been revised as "Their concentration time series exhibited moderate agreement, suggesting potential land-sea interactions." in Line 420-421.

4.5 Line 322: Requires a reference.

Response: We have incorporated supporting reference "which was also the major anthropogenic source of TGM (Pacyna et al., 2006; Streets et al., 2011; Liu et al., 2019)." in Line 424-425.

4.6 Lines 426–428: Wouldn't higher temperatures enhance the partitioning of Hg° from water to air? The authors should clarify this mechanism.

Response: Thank you for the comment. We agree with you that higher temperatures enhance the partitioning of Hg$^0$ from water to air, as Hg$^0$ inherently possesses high volatility. Elevated temperature intensifies molecular thermal motion, thereby accelerating the volatilization of mercury from the liquid phase to the gas phase.

In the revision (Line 557-560), we have rephrased this sentence as "Higher temperature not only favored the production of DGM in seawater (Costa and Liss, 1999; Andersson et al., 2011; Mason et al., 2001) but also promoted the escape of DGM from the water surface into the atmosphere (Osterwalder et al., 2021; Huang and Zhang, 2021)."

4.7 Lines 427–428: Supporting evidence or references are needed for the statement.

Response: The production of Hg$^0$ in seawater primarily originates from the photochemical reduction of divalent mercury (Hg$^{2+}$) (Costa and Liss, 1999; Andersson et al., 2011), which is closely linked to parameters such as light intensity and temperature (Ci et al., 2016; Mason et al., 2001). Both observational and modeling studies demonstrate that rising temperatures enhance oceanic mercury release. For example: Observations in the Baltic Sea revealed that seawater Hg$^0$ concentrations and air-sea exchange fluxes peak at midday, aligning with temperature fluctuations (Osterwalder et al., 2021). Model simulations indicate that uniformly raising sea surface temperature (SST) by 1 °C increases global Hg$^0$ evasion, particularly in high-latitude regions (1–8%), where relatively low SST originally limits Hg$^0$ evasion (Huang and Zhang, 2021).

We have revised the explanation in the updated manuscript and cited relevant literature. Specifically, the original statement "Higher temperature was generally more favorable for the formation of DGM" has been rephrased as "Higher temperatures not only favor the production of DGM in seawater (Costa and Liss, 1999; Andersson et al., 2011; Mason et al., 2001) but also

promote the escape of DGM from the water surface into the atmosphere (Osterwalder et al., 2021; Huang and Zhang, 2021)." in Line 557-560.

4.8 Lines 430–431: Requires citations.

Response: We have now added relevant references to this statement as below (Line 564).

"This suggested that continental inputs, such as river discharge, had a significant influence on DGM levels in nearshore waters (Chen et al., 2020; Kuss et al., 2018; Liu et al., 2016)."

5. Insufficient methodological details. For the ancillary data of ion concentrations, trace gases, and meteorological variables, the authors provided little information on the instruments used, and no information on data quality control and assurance as well as temporal resolution. Also, where were the PBL data from? They were introduced abruptly at one point in the results section.

Response: Thanks for the suggestion. We have added detailed methodological descriptions in Section 2 (Line 219-262).
At JHI, water-soluble ions in $PM_{2.5}$, including sulphate ($SO_4^{2-}$), nitrate ($NO_3^-$), ammonium ($NH_4^+$), chloride ($Cl^-$), sodium ($Na^+$), potassium ($K^+$), magnesium ($Mg^{2+}$), calcium ($Ca^{2+}$), alongside the soluble gases such as ammonia ($NH_3$) and sulfur dioxide ($SO_2$) were continuously monitored using an In-situ Gas and Aerosol Composition monitoring system (IGAC) (Wang et al., 2022). IGAC operated at a 1-hour temporal resolution and consisted of a wet annular denuder (WAD) and ion chromatography (IC) equipped with columns CS17 and CG17 for cations and AG11-HC and AS11-HC for anions. Ambient air was drawn into a $PM_{2.5}$ cyclone inlet by a built-in pump at a flow rate of 16.7 L/min. The sampled air was separated by passing through the vertically placed WAD to capture water-soluble gases, and airborne particles were collected by a steam scrubber and impact aerosol collector placed downstream. Air samples were dissolved by 30 ml ultra-pure water (18.25 MΩ $cm^{-1}$) and then divided into two steams. Both aqueous samples (including particles and gases) were injected into the IC system by two separated syringe pumps for analyzing the cations and anions. For quality assurance/quality control (QA/QC) of IGAC, a standardized lithium bromide (LiBr) solution was continuously introduced into aerosol liquid samples during the campaign to validate sampling and analytical stability. Weekly calibrations were performed for the ion chromatography (IC) module using certified standard solutions, with linearity ($R^2 > 0.99$) and detection limits (LODs) validated. Black carbon (BC) in $PM_{2.5}$ was measured continuously using a multi-wavelength Aethalometer (AE-33, Magee Scientific, USA). Meteorological parameters were measured using a Vaisala WXT530 surface weather station (Vaisala, Finland). Surface seawater temperature was recorded by a YSI EC300 portable conductivity meter (YSI, USA) with a resolution of 0.1°C.

At HNI, methods for analyzing meteorological parameters, BC, and surface seawater temperature mirrored those employed at JHI.

During the cruise campaign, the meteorological metrics (e.g., air temperature, wind speed/direction) and surface seawater temperature were collected from the Finnish Vaisala AWS430 shipborne weather station onboard the R/V. AE-33 was also used for BC measurements during the cruise.

At DSL, water-soluble ions in PM$_{2.5}$ and soluble gases were also measured by the IGAC instrument. Trace metals in PM$_{2.5}$ (Al, Ti, V, Cr, Mn, Fe, Co, Ni, Cu, Zn, Ga, As, Sr, Cd, Sn, Sb, Ba, Tl, Pb, and Bi) were continuously measured using an Xact multi-metals monitor (Model Xact™ 625, Cooper Environmental Services LLT, OR, USA). It operated at a flow rate of 16.7 L min$^{-1}$ with hourly resolution. Particles in the airflow passed through a PM2.5 cyclone inlet and were deposited onto a Teflon filter tape, then the samples were transported into a spectrometer for analysis via nondestructive energy-dispersive X-ray fluorescence.

Planetary boundary layer (PBL) height data were obtained from the Global Data Assimilation System (GDAS) archive maintained by the U.S. National Oceanic and Atmospheric Administration (NOAA), available through the READY (Real-time Environmental Applications and Display sYstem) portal (https://www.ready.noaa.gov/archives.php; last accessed: 11 May 2025). The dataset, featuring 1-hour temporal resolution, was processed and extracted using MATLAB R2021b (MathWorks, Natick, MA).

6. Random citations. Some references seemed to be cited arbitrarily. While citing every study on a given topic is impractical, it is important to acknowledge milestone research appropriately. Here are a few examples. There have been hundreds and thousands of journal articles on PMF applications. Did Qin et al. (2020) develop the PMF approach? Was Gibson et al. (2015) the first to recognize PMF "for its efficacy in elucidating sources profiles and quantifying source contributions"? In lines 280-281, were those studies the first to establish the role of temperature in GEM evasion? In lines 321-322, were those studies the first to identify fossil fuel combustion as a major mercury source?

6.1 There have been hundreds and thousands of journal articles on PMF applications. Did Qin et al. (2020) develop the PMF approach? Was Gibson et al. (2015) the first to recognize PMF "for its efficacy in elucidating sources profiles and quantifying source contributions"?

Response: Thank you for the comments. The original statement "Qin et al. (2020) develop the PMF approach" has been revised, and "Gibson et al. (2015)" is no longer cited as the first to recognize PMF "for its efficacy in elucidating source profiles and quantifying source contributions." The theoretical foundation of the PMF model was originally documented in the study by Paatero and Tapper (Paatero and Tapper, 1994). In the revised manuscript, the citations in lines 267-288 have been modified as follows.

- "The PMF model is recognized for its efficacy in elucidating source profiles and quantifying source contributions (Paatero and Tapper, 1994)."

- "Detailed descriptions can be found in previous study (Paatero and Tapper, 1994)."

6.2 In lines 280-281, were those studies the first to establish the role of temperature in GEM evasion?

Response: The role of temperature in mercury evasion was indeed established earlier. Temperature-driven Hg⁰ evasion has been mechanistically characterized since the 1990s (Lindberg et al., 1998; Poissant et al., 2000). We have updated the references to prioritize pioneering studies.

6.3 In lines 321-322, were those studies the first to identify fossil fuel combustion as a major mercury source?

Response: The original citation overlooked earlier work. The text now cites the early global mercury emission inventory identifying fossil fuels as a dominant anthropogenic source (Pacyna et al., 2006; Streets et al., 2011).

7. Uncertainty in sea-air exchange flux calculations using TGM as a proxy for Hg° in sea-air exchange flux calculations could introduce significant uncertainty. While this may be reasonable in a landlocked atmosphere, it can be problematic in the marine boundary layer, where halogen compounds are abundant and subsequently GOM concentrations are probably not negligible at times. For example, Castagna et al. (2018, atmos. Env.) reported GOM reaching well over 100 pg/m3, ~10% of TGM, at times. Note that in the reference cited, GOM was measured using the Tekran series, which has been in the literature suggested to be largely under-biased, primarily by Gustin et al.'s team. The actual GOM concentrations may be even higher.

Response: Thanks for your insightful comments. As noted, Gustin et al. (2015) highlighted that GOM concentrations measured by KCl-coated denuders could be underestimated by a factor of 2 to 3. To investigate the impact of this GOM measurement underestimation on the calculation of air-sea exchange fluxes in our study, we compiled observed GEM and GOM concentrations in China's marginal seas (listed in the table below). We found that even after accounting for the 2- to 3-fold GOM underestimation reported by Gustin et al. (2015), the proportion of GOM relative to TGM (GOM + GEM) in China's marginal seas didn't exceed 1.2%. Therefore, we conclude that the error introduced by substituting TGM for GEM in calculating air-sea exchange fluxes should be negligible in the China's marginal seas.

Table. GEM and GOM concentrations in China's marginal seas

| Location | Site description | Sampling time | GEM (ng/m³) | GOM (pg/m³) | Reference |
|---|---|---|---|---|---|
| China | South China Sea | 2015 | 1.52±0.32 | 6.1±5.8 | (Wang et al., 2019) |
| | Bohai Sea and Yellow Sea | 2014 (spring) | 2.03±0.72 | 2.5±1.7 | (Wang et al., 2016b) |
| | Bohai Sea and Yellow Sea | 2014 (fall) | 2.09±1.58 | 4.3±2.5 | (Wang et al., 2016b) |
| | East China Sea (HNI) | 2013-2014 | 2.25 ± 1.03 | 8 ± 10 | (Fu et al., 2018) |

Andersson, M. E., Sommar, J., Gårdfeldt, K., and Jutterström, S.: Air–sea exchange of volatile mercury in the North Atlantic Ocean, Mar. Chem., 125, 1-7, https://doi.org/10.1016/j.marchem.2011.01.005, 2011.

Chang, Y. H., Huang, K., Xie, M. J., Deng, C. R., Zou, Z., Liu, S. D., and Zhang, Y. L.: First long-term and near real-time measurement of trace elements in China's urban atmosphere: temporal variability, source apportionment and precipitation effect, Atmospheric Chemistry and Physics, 18, 11793-11812, 10.5194/acp-18-11793-2018, 2018.

Chen, Y.-S., Tseng, C.-M., and Reinfelder, J. R.: Spatiotemporal Variations in Dissolved Elemental Mercury in the River-Dominated and Monsoon-Influenced East China Sea: Drivers, Budgets, and Implications, Environmental science & technology, 54, 3988-3995, 10.1021/acs.est.9b06092, 2020.

Ci, Z., Zhang, X., Yin, Y., Chen, J., and Wang, S.: Mercury Redox Chemistry in Waters of the Eastern Asian Seas: From Polluted Coast to Clean Open Ocean, Environmental science & technology, 50, 2371-2380, 10.1021/acs.est.5b05372, 2016.

[revised manuscript text omitted]

---

## Author Comment (AC2)

**Response to Reviewer #3's Comments**

Anonymous Referee #3:

Major comments:

The study by Qin et al. investigates the atmospheric TGM concentrations at two islands and over the coastal oceans in east Asia, and explores the effect and contributions of anthropogenic emissions based on atmospheric tracer ratios and a receptor model. The finding from this study is valuable for understanding the cycling of atmospheric Hg in coastal ocean regions. This study provides many observational data, and does an insightful analysis of the datasets. The manuscript is well written and I broadly agree with the interpretations of the new data. I think the manuscript is currently in good quality. I have provided a number of minor comments that hope to be considered by the authors.

We sincerely thank for your in-depth comments and helpful suggestions on this manuscript. Based on the specific comments, we have responded to all the comments point-by-point and made corresponding changes in the manuscript as highlighted in the track change mode. You have raised a number of issues and we quite agree. We feel the substantial revisions based on your comments have greatly improved the quality of this manuscript. Please check the detailed responses to all the comments as below.

Specific comments:

1. the abstract: the levels and distribution patterns of atmospheric TGM are one of the major contributions of this study, which are better to be briefly summarized in the abstract.

Response: Thanks for your suggestion. The abstract has been revised to emphasize the distribution patterns and levels of TGM. The following sentence was added in Line 32-34.

"The mean concentrations of TGM were 2.32 ± 1.02 ng/m³ (Bohai Sea), 2.55 ± 0.55 ng/m³ (Yellow Sea), and 2.31 ± 0.81 ng/m³ (East China Sea), respectively, with coastal regions exhibiting significantly higher values than open ocean areas due to continental outflows."

2. line 39: these numbers should be linked to the seas investigated in this study.

Response: The text has been clarified to explicitly associate each percentage with its corresponding sea in Line 44-47.

"To assess the potential impact of anthropogenic emissions on the sea-air exchange fluxes of mercury, anthropogenic contributions to TGM were artificially removed, then the fluxes would be increased by 207.1% in the Bohai Sea, 33.4% in the Yellow Sea, and 6.5% in the East China Sea, respectively."

3. line 228: better to specify the equations for the calculation of Schmidt number of elemental mercury. In addition, provide the methods for detection of wind speed and water temperature.

Response: Thanks for the suggestion. The Schmidt number calculation and instrumentation details have been added in the revision (Line 304-308).

"The Schmidt number for Hg ($S_{C_{Hg}}$) was calculated as:

$$S_{C_{Hg}} = v/D_{Hg}$$

where $v$ is seawater kinematic viscosity (Wanninkhof, 2014) and $D_{Hg}$ is the diffusion coefficient of Hg (Kuss et al., 2009)."

The wind speed measurements at Huaniao Island and Juehua Island employed the Vaisala WXT530 surface weather station (Vaisala, Finland) with a resolution of 0.01 m/s. The surface seawater temperature was measured using a YSI EC300 portable conductivity meter (YSI, USA) with a resolution of 0.1°C. During the voyage, wind speed and surface seawater temperature data were acquired by the Finnish Vaisala AWS430 shipborne weather station with resolutions of 0.01 m/s and 0.1°C, respectively. The information has been added in Section 2.4.

4. line 235-239 and 247-248: this study describes well the spatial and temporal distribution patterns. As far as I have concerned, recent long-term continuous observations of TGM in mainland China have already showed significant declines in the past decades (e.g., Feng et al., 2024, NSR). My question is whether long-term changes in TGM in Chinese coastal oceans is similar to the mainland. I therefore suggest the author compare with previous observations and show the trends of TGM in coastal oceans in east Asia.

Response: Thanks for your valuable suggestions. The following table compares the GEM/TGM concentrations observed in this study and previous research in the offshore areas of East Asia. We found that existing observations struggle to clarify the long-term trends of TGM in marine environments, primarily due to two reasons: 1) Insufficient observational frequency—for example, previous studies on Bohai Sea only reported TGM data from spring and autumn of 2014; 2) Difficulty in comparing TGM concentrations across different research cruises, as sampling locations, timing, and cruise durations (typically spanning only a few weeks) varied significantly among different studies.

To infer trends in marine TGM concentrations, long-term monitoring data from coastal cities could serve as a reference. For instance, one study at Chongming Island near the coastal East China Sea revealed a marked decline in GEM concentrations from 2014 to 2016, with a rate of -0.60 ± 0.08 ng m$^{-3}$ yr$^{-1}$; Research at Dianshan Lake, Shanghai, also indicated a decline rate of

-0.32 ± 0.07 ng m$^{-3}$ yr$^{-1}$ for GEM concentrations from 2015 to 2018. These findings suggest a potential downward trend in TGM concentrations across the offshore regions of East Asia.

Table. The GEM/TGM concentrations in this study and other literature.

| Location | Site description | Sampling time | TGM/GEM (ng/m$^3$) | Reference |
|---|---|---|---|---|
| BS | Nearshore islands | 2020 (winter) | 2.32 ± 1.02 | This study |
| YS | Sea | 2020 (winter) | 2.55 ± 0.55 | This study |
| ECS | Sea | 2020 (winter) | 2.31 ± 0.81 | This study |
| ECS | Sea(Huaniao island) | 2020(autumn) | 1.85 ± 0.74 | This study |
| BS | Sea | 2014 (spring) | 2.51 ± 0.77 | Wang et al. (2016a) |
| BS | Sea | 2014 (autumn) | 3.64 ± 2.54 | Wang et al. (2016a) |
| YS | Sea | 2014 (spring) | 1.89 ± 0.64 | Wang et al. (2016a) |
| YS | Sea | 2014 (autumn) | 1.59 ± 0.44 | Wang et al. (2016a) |
| YS | Sea | 2012(spring) | 1.86±0.40 | Ci et al. (2015) |
| YS | Sea | 2012(autumn) | 1.84±0.50 | Ci et al. (2015) |
| YS | Sea | 2010(summer) | 2.61±0.50 | Ci et al. (2011) |
| ECS | Sea (Huaniao island) | October 2013 to January 2014 | 2.25 ± 1.03 | Fu et al. (2018) |
| ECS | Sea | 2014 (spring) | 1.61 ± 0.32 | Wang et al. (2016a) |
| ECS | Sea | 2013(summer) | 1.61 ± 0.32 | Wang et al. (2016b) |
| ECS | Sea | 2013(autumn) | 2.20 ± 0.58 | Wang et al. (2016a) |
| SCS | Sea | 2015(autumn) | 1.52±0.32 | Wang et al. (2019) |
| SCS | Sea | 2007(summer) | 2.62±1.13 | Fu et al. (2010) |

5. line 261-263: this statement is very speculative and in contrast with the discussions in sections below. The good relationship observed could be affected by many factors. Generally, ocean Hg emissions are largely controlled by wind speed, while the effect of temperature and solar radiations is relatively small.

Response: Thank you for your comments. We agree with the reviewer that wind speed and wind turbulence are the most direct factors influencing oceanic mercury release, which can be reflected by the discussions related to Figure 3. Accordingly, we have revised the statement as "The TGM diurnal pattern displayed strong concordance with temperature and solar flux (Figure 2a)." in Line 348-350.

6. line 280-281: have any of previous studies provided solid evidence that increasing temperature would facilitate strong ocean Hg emissions?

Response: Thanks for your comments. The air-sea exchange of $Hg^0$ is driven by concentration gradients across the atmospheric and seawater interface (Soerensen et al., 2013). From a physical perspective, $Hg^0$ inherently possesses high volatility. Elevated temperatures intensify molecular thermal motion, accelerating the volatilization of mercury from the liquid phase to the gas phase, thereby increasing the risk of oceanic mercury release. This is reflected in the calculation of mercury's air-sea exchange flux. According to the two-film theory model (Wanninkhof and Oceans, 1992), the air-sea exchange flux of mercury is calculated as:

$$F = K_W(C_W - C_a/H')$$

where Kw represents the gas exchange velocity. Kw is determined by the formula (Nightingale et al., 2000):

$$K_W = 0.31u_{10}^2(S_{C_{Hg}}/660)^{-0.5}$$

where $S_{C_{Hg}}$ is the Schmidt number of Hg0 in seawater, calculated as:

$$S_{C_{Hg}} = v/D_{Hg}$$

with D being the aqueous-phase diffusion coefficient of $Hg^0$, calculated as:

$$D = 0.0011e^{-(11.06\ kJmol^{-1})/RT}$$

Where T is the seawater temperature. Notably, under constant conditions, higher seawater temperatures lead to an increase in D, a decrease in the Schmidt number (Sc) of $Hg^0$ in seawater, and consequently an increase in the mass transfer coefficient Kw, resulting in higher air-sea exchange flux of mercury.

From a chemical perspective, the production of $Hg^0$ in seawater primarily originates from the photochemical reduction of divalent mercury ($Hg^{2+}$) (Costa and Liss, 1999; Andersson et al., 2011), which is closely linked to parameters such as light intensity and temperature (Ci et al., 2016; Mason et al., 2001). Both observational and modeling studies demonstrate that rising temperatures enhance oceanic mercury release. For example, observations in the Baltic Sea revealed that seawater $Hg^0$ concentrations and air-sea exchange fluxes peaked at midday, aligning with temperature fluctuations (Osterwalder et al., 2021). Model simulations indicated that uniformly raising sea surface temperature (SST) by 1 °C increased global $Hg^0$ evasion, particularly in high-latitude regions (1–8%), where relatively low SST originally limited $Hg^0$ evasion (Huang and Zhang, 2021).

In the revision, we have explained this in Line 557-560:

Higher temperature not only favored the production of DGM in seawater (Costa and Liss, 1999; Andersson et al., 2011; Mason et al., 2001) but also promoted the escape of DGM from the water surface into the atmosphere (Osterwalder et al., 2021; Huang and Zhang, 2021).

7. line 284-285: note that the natural emissions in this study is mainly associated with seawater emissions. while the discussions here is mainly reasonable for soil Hg emissions.

Response: Thanks for the comments. The references for this statement are now replaced with more relevant references that explicitly address mercury marine processes in Line 374-376.
 "Previous studies suggested wetting processes may promote the reduction of $Hg^{II}$ to $Hg^0$ in surface seawater, while higher wind speed accelerated its evasion (Lin et al., 2010; Soerensen et al., 2013)"

8. line 327-330: I thought the authors should determine the TGM/BC ratios based on the slope of the correlations between TGM and BC. It seems the ratios are calculated by observed levels of TGM and BC. Note that BC is not long-lived atmospheric pollutants and ready to deposit more quickly than TGM, and this is why high TGM/BC ratios was observed at locations far from sources.

Response: Thanks for the comments. We agree with the reviewer that BC is not a long-lived atmospheric pollutant and deposits more rapidly than TGM. The content in lines 327–330 originally aimed to discuss the differences in TGM/BC between terrestrial and marine environments from an observational perspective. However, the wording in the original conclusion was overly definitive. We have therefore revised the statement in Line 431-434.
"On one hand, lower contribution of anthropogenic sources to TGM in the coastal environment compared to the urban environment was expected. On the other hand, BC deposited more quickly than TGM, thus also elevating the TGM/BC ratios was at locations far from sources.".

9. line 370: it is interesting to provide quantitative analysis of the contributions of anthropogenic sources. My question is whether the results from this study agrees or is different from previous studies. Recently, several studies quantify the contributions of anthropogenic emissions to atmospheric TGM in rural areas in China (including HNI) based on Hg isotope approaches. I would therefor suggest the authors compare their results with previous isotope and modelling studies.

Response: Thanks for your valuable suggestions and we do agree that the results in this study should be compared to studies based on Hg isotope approaches. This study found that anthropogenic sources contributed 38%, 26%, and 35% to TGM in the Yellow Sea, East China Sea, and Huaniao Island, respectively. For Dianshan Lake in Shanghai and Juehua Island in Liaoning Province, the anthropogenic contributions rose to 47% and 59%, respectively. In comparison, previous isotope-based source apportionment studies have revealed anthropogenic contributions of 29% and 42% to TGM in remote areas like Changbai Mountain and Ailao Mountain (Wu et al., 2023). In general, the isotope-based results indicated that the relative contributions of anthropogenic emissions to surface GEM in remote China and urban China were around 30% and 49%, respectively (Fu et al., 2021; Feng et al., 2022; Wu et al., 2023). Notably, the anthropogenic contributions to TGM in the Yellow Sea, East China Sea, and Huaniao Island

from this study align closely with isotope-derived values from China's remote regions, while the Dianshan Lake findings correspond with urban isotope results. The elevated contribution observed at Juehua Island (59%) may be attributed to its proximity to the mainland (only 10 km away) and the sampling period occurring during the winter heating season, where continental transport influences were significant (Li et al., 2023). Furthermore, the values obtained in this study fall within comparable ranges to modeling study estimates (typically 33% to 41% on average)(Chen et al., 2014; Wang et al., 2018).

The comparison between this study and previous studies have been added in Line 524-536.

10. line 381-384: see my previous comments on TGM/BC ratios. Are there any strong relationship between TGM and BC over the seas? If yes, better to use relationship slopes to estimate the contributions of anthropogenic emission over the seas, but not using the island results.

Response: Thanks for your valuable comments. We have now checked the relationship between TGM and BC over the seas. As shown in the figure below, the concentrations of TGM and BC observed over the ocean in this study did not exhibit a strong correlation. This was indeed anticipated, as TGM was influenced not only by anthropogenic sources but also significantly by natural sources, particularly over the ocean, while BC primarily originated from anthropogenic emissions. In addition, the cruise observation covered a wide oceanic area. Different oceanic regions had different relationship between natural vs. anthropogenic mercury. Thus, we used other approaches to quantify the contributions of anthropogenic emission over the seas in this study.

[Figure]

Figure. Scatter plot of TGM and BC concentrations over the ocean and their correlation

11. Line 422-423: better to show the values or ranges of previously observed DGM in the seas.
Response: Thanks for your suggestions. We have now added the concentration ranges of DGM from previous literature in Line 549-551.
"The DGM concentrations measured during this winter cruise campaign (22.9-39.7 pg/L) were significantly lower than those recorded previously during summer and fall in similar regions (52.4-63.9 pg/L) (Ci et al., 2011; Ci et al., 2015; Wang et al.,2016a)."

12. Line 440-441: I would suggest the authors to provide detailed information regarding the DGM and air TGM concentrations, wind speed, water temperature, and calculated exchange flux at each of the sampling sites in the supporting information. This would be valuable for future studies in air-sea Hg exchanges.
Response: Thanks for your suggestions. We have added a new table in the supplementary materials, which includes parameters such as DGM and TGM concentrations, wind speed, water temperature, and exchange flux, as shown in the table below.

Table S1. DGM and TGM concentrations, wind speed, water temperature, and sea-air flux at each sampling site during the cruise campaign.

[revised manuscript text omitted]

---

## Author Comment (AC3)

**Response to Reviewer #2's Comments**

Anonymous Referee #2:

Major comments:

In this study, the authors investigated the contributions of anthropogenic and natural sources to atmospheric mercury in the marine environment in eastern China based on island, cruise, and inland campaigns. Correlation analyses were performed between TGM and meteorological factors. The TGM/BC ratios were calculated. PMF models were developed for two sites to disentangle anthropogenic and natural contributions, and correlations between BC and anthropogenic TGM were established. The sea-air exchange fluxes of mercury were estimated based on TGM and DGM. The impacts of anthropogenic emissions on marine Hg evasion were quantified. Overall, the study is well designed, and results from this study provide more evidence for the contribution of anthropogenic sources to TGM in the marine boundary layer in eastern China and the compensation effect of marine Hg evasion when the anthropogenic contribution is reduced. However, the methods part lacks some detailed information, and the results and discussion part needs improvement to address the novelty of this study. There are many speculative statements in the discussion part, the expression of which could be improved. Therefore, in my opinion, major revision is required before the manuscript is acceptable for publication on Atmospheric Chemistry and Physics.

We sincerely thank for your in-depth comments and helpful suggestions on this manuscript. Based on the specific comments, we have responded to all the comments point-by-point and made corresponding changes in the manuscript as highlighted in the track change mode. You have raised a number of issues and we quite agree. We feel the substantial revisions based on your comments have greatly improved the quality of this manuscript. Please check the detailed responses to all the comments as below.

Specific comments:

1. Line 46: Typo of "can cycles".

Response: Thank you for pointing out this error. We have corrected "can cycles" to "can cycle".

2. Lines 60–71: The statements of these literatures are inadequate. For example, Fu et al. (2018) and Wang et al. (2020) are not considered as qualitative assessment. They have provided quantitative evidence for the contributions of anthropogenic Hg emissions. I suggest these statements be rephrased.

Response: Thanks for your comments. In the revised manuscript, we have modified the original statement as below.

Line 79-85: "Although isotopic signatures have been widely applied to source apportionment of atmospheric mercury, current isotopic methods still exhibit significant uncertainties due to the poor understanding of isotopic compositions of gaseous elemental mercury emitted from various sources and fractionation processes of Hg isotopes during atmospheric transformations

(Fu et al., 2018). Additionally, this approach requires specialized isotopic measurements unavailable for routine monitoring. At present, quantitative analyses of anthropogenic contributions to marine atmospheric mercury remain limited."

3. Section 2.2: The QA/QC results for TGM/GEM measurement should be further illustrated. For example, what are the average duplication rates between A and B traps for the Tekran 2537B and the modified 2600 analyzer, respectively? What measures did the authors take to prevent or abate the impacts of high humidity on the Tekran analyzers?

Response: Thanks for your comments. For the Tekran 2537B, the average duplication rate between the A and B traps is 99%, with deviations between the two traps consistently below 3%. To mitigate the impact of high humidity on the instrument, samples are first passed through a soda lime drying tube for dehumidification before entering the detector.

The modified Tekran 2600 adopts a simplified design without separating A and B traps. Instead, it utilizes two gold traps for mercury collection and analysis. During operation, atmospheric mercury is adsorbed onto the first gold trap over a 24-minute sampling period. After sampling, the mercury on the first gold trap is thermally desorbed and transferred to the second gold trap. The second trap is then analyzed by the detector during a 6-minute detection phase, resulting in an overall 30-minute sample resolution. To ensure data quality during cruise observations, the instrument is calibrated daily using the external calibration unit Tekran 2505. Similar to the 2537B, samples are also pre-dried via a soda lime drying tube prior to detector entry to prevent humidity interference.

In the revised manuscript, we have added the following contents to Section 2.2.

Line 190-193: For the Tekran 2537B, the average duplication rate between the A and B traps is 99%, with deviations between the two traps consistently below 3%. To mitigate the impact of high humidity on the instrument, samples are first passed through a soda lime drying tube for dehumidification before entering the detector.

Line 166-173: During the operation of the modified Tekran 2600, atmospheric mercury was adsorbed onto the first gold trap over a 24-minute sampling period. After sampling, the mercury on the first gold trap was thermally desorbed and transferred to the second gold trap. The second trap was then analyzed by the detector during a 6-minute detection phase, resulting in an overall 30-minute sample resolution. To ensure data quality during cruise observations, the instrument was calibrated daily using the external calibration unit Tekran 2505. Samples were pre-dried via a soda lime drying tube prior to detector entry to prevent humidity interference.

4. Section 2.5: More illustration on the PMF method need to be included instead of just referring to the authors' previous study. For example, at least what indicators were used in the PMF model should be introduced. According to Section 3.3, air temperature (I assume it is air temperature instead of seawater temperature) has been included in the model. What unit did the authors use? I think Kelvin makes more sense than °C since the indicators need to be positive for applying PMF.

Response: Thanks for your comments. The original description of the PMF model was indeed simplified. In the revised manuscript, we have addressed this by:

1. Adding the formula for calculating the Q-value (the objective function) in Section 2.5, along with a detailed explanation of its significance.
2. Elaborating on the criteria for selecting the number of factors (tested from 3 to 8) based on the slope of the Q-value versus the number of factors.

The updated text in Section 2.5 now reads (Line 274-292):

"The objective function, defined in Eq. (2) below, represents the sum of the squared differences between measured and modeled concentrations, weighted by concentration uncertainties. Minimizing this function allows the PMF model to determine optimal non-negative factor profiles and contributions:

$$Q = \sum_{i=1}^{n} \sum_{j=1}^{m} \left( \frac{X_{ij} - \sum_{k=1}^{p} A_{ik}F_{kj}}{S_{ij}} \right)^2$$

Where $X_{ij}$ denotes the concentration of the $j^{th}$ pollutant in the $i^{th}$ sample, $A_{ik}$ represents the contribution of the $k^{th}$ factor to the $i^{th}$ sample, $F_{kj}$ is the mass fraction of the $j^{th}$ pollutant in the $j^{th}$ pollutant in $k^{th}$ factor, $S_{ij}$ is the uncertainty of the $j^{th}$ pollutant in the $i^{th}$ sample, and p is the number of factors.

TGM, air temperature (unit: Kelvin), gaseous pollutants, and major aerosol chemical species were used as inputs for the PMF model. We tested factor numbers ranging from 3 to 8, with the optimal solution determined by analyzing the slope of the Q-value versus factor count. Model stability was assessed through residual analysis, correlation coefficients between observed and predicted concentrations, and Q-value trends. A six-factor solution in DSL and a five-factor solution at JHI provided the most stable and interpretable results.

For the air temperature input, we confirm that values were converted to Kelvin (°C + 273.15) prior to PMF analysis. This aligns with the non-negativity constraint inherent to PMF's mathematical framework, as rightly noted by the reviewer.

5. Lines 249–250: The reference here for background level in Northern Hemisphere is outdated. Please refer to more recent studies (e.g., Bencardino et al., 2024). It is also encouraged to give the value range in the text.

Response: Thanks for your comments. Following updates from recent literature, this statement in the revised manuscript has been modified in Line 327-330.

"The mean TGM concentrations reached 2.36 ± 0.65 ng/m³ and 2.16 ± 0.81 ng/m³ over the Yellow Sea and East China Sea, respectively, significantly higher than the background level in the Northern Hemisphere (1.58 ± 0.31 ng/m³) (Bencardino et al., 2024).".

6. Lines 250–253: Similarly, it is recommended to list the mean values and standard deviations of TGM concentrations in these studies.

Response: Following your suggestion, we have listed the mean values and standard deviations of TGM concentrations from comparative studies. This sentence is revised in Line 332-335.

"also surpassing measurements recorded in the other open ocean areas such as the South China Sea (1.52±0.32 ng/m³), Mediterranean Sea (1.8± 1.0 ng/m³), Bering Sea (1.1 ± 0.3 ng/m³), Pacific Ocean (1.15-1.32ng/m³), and Atlantic Ocean (1.63 ± 0.08 ng/m³)".

7. Lines 331–333: These TGM/BC ratio ranges are a bit strange. These values were

automatically generated on mapping. I suggest the authors reset the ranges and use rounded values instead.

Response: Thanks for your suggestion. We have now adjusted the TGM/BC ratio ranges. The revised Figure 4 (see below) now reflects these modifications.

[Figure]

8. Lines 363–370: The method for measuring the concentrations of heavy metals at DSL was not mentioned in Section 2. Was it an online or offline method? How big was the dataset?

Response: In the revised manuscript, we have introduced the online methodology for measuring heavy metals in Section 2. The data used in this study covers the period from October to December, 2020 with a temporal resolution of 1 hour.

The details are added in Line 251-256.

"Trace metals in $PM_{2.5}$ (Al, Ti, V, Cr, Mn, Fe, Co, Ni, Cu, Zn, Ga, As, Sr, Cd, Sn, Sb, Ba, Tl, Pb, and Bi)were continuously measured using an Xact multi-metals monitor (Model Xact™ 625, Cooper Environmental Services LLT, OR, USA). It operated at a flow rate of 16.7 L $min^{-1}$ with hourly resolution. Particles in the airflow passed through a PM2.5 cyclone inlet and were deposited onto a Teflon filter tape, then the samples were transported into a spectrometer for analysis via nondestructive energy-dispersive X-ray fluorescence."

9. Figure 5 (c and d): It seems to me that the two relationships are quite similar. The relationship for JHI is highly influenced by the top right data point, which is not robust. Therefore, I suggest the authors integrate the data points of these two sites and establish a uniform relationship.

Response: Thanks for your comments. The relationships between anthropogenic GEM and BC were originally derived from the daily average data. Now, we have used the original hourly data to re-calculate the linear regressions, which are shown in the figure below. It could be seen that even there is a number of high concentration data on the top right, the regression equations are close to the original relationships based on daily data. This indicates that the relationship between anthropogenic GEM and BC is still robust.

[Figure]

[Figure]

We appreciate for your suggestion that a uniform relationship can be established by integrating the data points of two sites. However, it could be seen that regression slopes at the two sites have discernible differences. This was reasonable as northern China and southern China have different energy structures, thus the relationship between anthropogenic GEM and BC should be different to some extents. In this regard, we didn't use the uniform relationship for the further data analysis.

10. Figure 6 (a-d): These relationships are all based on nonlinear regressions which are not consistent with the linear assumption for PMF.

Response: Thank you for the comments. The linearity assumption of PMF applies strictly to the additivity of factor contributions but does not constrain the intrinsic relationships among input variables. As explicitly stated in the US EPA PMF 5.0 Guide, input variables may include nonlinearly correlated features (e.g., meteorological parameters) provided their data uncertainties are properly quantified. When temperature is included in PMF as a continuous variable, its association with pollution sources may be linear or nonlinear, which does not violate the model's core assumptions. The nonlinear regressions in Figure 6 are independent of the PMF factorization process; their sole purpose is to reveal potential nonlinear modulation effects of temperature on the intensity of source contributions, serving as descriptive analyses that do not participate in PMF model computations. Therefore, the nonlinear relationships shown in Figure 6 are methodologically justified.

11. Lines 411–415: Could it be more likely that the Yellow Sea suffers more from the air masses from the North China Plain region which is more polluted than eastern China?

Response: Thank you for the comment. As shown in Figure S1, the backward trajectories during the Yellow Sea segment of the cruise predominantly originated from the North China Plain and Liaoning Province. We therefore agree with your perspective that "the Yellow Sea suffers more from the air masses from the North China Plain region, which is more polluted than eastern China".

In the revised manuscript, we have revised this description in Line 536-539.

"As shown in Figure S1, the backward trajectories over the Yellow Sea segment were primarily influenced by air masses from the North China Plain and Liaoning Province. The relatively

higher contribution of anthropogenic sources to the Yellow Sea during the cruise was likely attributable to the continental transport from northern China."

[Figure]

Bencardino, M., D'Amore, F., Angot, H., Angiuli, L., Bertrand, Y., Cairns, W., Diéguez, M. C., Dommergue, A., Ebinghaus, R., Esposito, G., Komínková, K., Labuschagne, C., Mannarino, V., Martin, L., Martino, M., Neves, L. M., Mashyanov, N., Magand, O., Nelson, P., Norstrom, C., Read, K., Sholupov, S., Skov, H., Tassone, A., Vítková, G., Cinnirella, S., Sprovieri, F., and Pirrone, N.: Patterns and trends of atmospheric mercury in the GMOS network: Insights based on a decade of measurements, Environmental Pollution, 363, 125104, https://doi.org/10.1016/j.envpol.2024.125104, 2024.

Fu, X., Yang, X., Tan, Q., Ming, L., Lin, T., Lin, C.-J., Li, X., and Feng, X.: Isotopic Composition of Gaseous Elemental Mercury in the Marine Boundary Layer of East China Sea, Journal of Geophysical Research: Atmospheres, 10.1029/2018jd028671, 2018.

---

## Author Response (AR2)

**Response to Reviewer #1's Comments**

The authors have addressed my questions and comments about their PMF application and a few specific comments.

1. However, I have a few remaining concerns. First, I don't find their response convincing, especially to one of the main questions I had: why did they estimate the oceanic evasion flux of GEM after removing the anthropogenic contribution? This estimation is after all the crux of the study. They replied that "(t)he purpose of this recalculation is to assess potential changes in the marine mercury flux under the scenario of reduced anthropogenic emissions". But does this imply that anthropogenic emissions should not be reduced simply because oceanic evasion might increase as a result? According to Eq. 1, removing all the anthropogenic contribution in Ca, while assuming that Cw remains unchanged, increases the concentration gradient and thereby inflates the estimated evasion flux. The issue is whether Cw would in fact remain constant if all anthropogenic contributions were eliminated. I suspect not—particularly in near-coastal regions, where reduced Ca would likely lead to less dry deposition to the ocean surface, and thus a decrease in Cw. Ignoring this positive feedback between anthropogenic emissions and oceanic evasion is, in my view, a major flaw in the study.

Response: Thank you for your insightful comments. We agree with the reviewer's opinion that it is unreasonable to recalculate the sea-air exchange flux by directly deducting the anthropogenic contribution. In the original submission, we aimed to evaluate the instantaneous mercury flux by changing the inputs into the model. However, we also realize that it will be a long-term process by cutting off all the anthropogenic mercury emissions. Thus, the Cw concentrations should gradually change along with the decreases of anthropogenic mercury emissions as you commented. In this regard, the variation of Cw under the scenario of anthropogenic mercury emissions reduction is hard to predict.

In the revised manuscript, we have removed the inappropriate discussions in Line 581-591 and added some discussions about the mercury sea-air exchange observed in different oceanic regions.

Line 572-577: This finding coincided with the discussions above that natural TGM exhibited much higher concentrations in the ECS and YS than in the BS.

Due to the higher concentrations and contributions of anthropogenic TGM in the BS, the release of mercury from the ocean was significantly suppressed, which likely explained the relatively low sea-air exchange flux of mercury there.

2. Second, the authors simply removed the TGM/BC ratio originally presented in the abstract in response to my question regarding its significance. However, the relevant content remains in the manuscript without further consideration. Even the original values were not updated despite the use of a different dataset and recalculated values (See Lines 428-436). BC concentrations

naturally decreased over the ocean as the distance from the coast increased. However, TGM concentrations are more complex, particularly given the authors' own finding of a significant oceanic source for GEM. The pattern of TGM/BC varied between region south and north of ~32.5 °N: south of this latitude, the ratio increases with increasing distances from the coast, whereas the pattern to the north is more complicated. This suggests that the TGM/BC ratio is rather confounding, doesn't it? I have no objection to including this result in the manuscript, but the authors must provide a more meaningful discussion of its implications, which is currently lacking.

2.1 Even the original values were not updated despite the use of a different dataset and recalculated values (See Lines 428-436).

Response: Thank you for your comments. As you mentioned that "Even the original values were not updated despite the use of a different dataset and recalculated values (See Lines 428-436).", this refers to the TGM/BC ratios observed at DSL and HNI as presented in Figure 4d. Actually, the TGM/BC ratios at these two sites were calculated based on the observational data from October 14 to November 4, 2020, i.e., the time-series shown in Figure 4a. In the previous revision, the data used in Figure 4 was not changed. We guess that you meant the dataset that was used for PMF was changed, however, that didn't influence the calculation of the TGM/BC ratios. Thus, the TGM/BC ratios remained unchanged.

2.2 BC concentrations naturally decreased over the ocean as the distance from the coast increased. However, TGM concentrations are more complex, particularly given the authors' own finding of a significant oceanic source for GEM. The pattern of TGM/BC varied between region south and north of ~32.5 °N: south of this latitude, the ratio increases with increasing distances from the coast, whereas the pattern to the north is more complicated. This suggests that the TGM/BC ratio is rather confounding, doesn't it? I have no objection to including this result in the manuscript, but the authors must provide a more meaningful discussion of its implications, which is currently lacking.

Response: We appreciate for your insightful comments and totally agree that the interpretation of the TGM/BC ratio indeed requires more in-depth discussions.

In Section 3.2, we aimed to emphasize the importance of marine emissions of TGM following Section 3.1. Ideally, the TGM/BC ratio should show an increasing trend with the increase of distances away from the coasts as the ocean emissions of TGM sustained while anthropogenic emissions over the ocean significantly diminished compared to the terrestrial areas. This could be verified by the significant difference of the TGM/BC ratio between DSL and HNI, even these two sites were only 150 km away. However, we only used the TGM/BC ratio as a qualitative index. That's why we further built a quantitative method to quantify the anthropogenic and natural contributions to TGM in the following sections.

We totally agree with you that the observed TGM/BC ratio exhibited distinct patterns in the East China Sea and the Yellow Sea. In the East China Sea, the TGM/BC ratio increased with distance from the coast, aligning with our expectations as anthropogenic influences diminished

farther offshore. However, the Yellow Sea showed a more complex situation, lacking the consistent trend observed in the East China Sea. This should be attributed to the Yellow Sea being a comparatively enclosed basin with more complex sources of TGM and BC, simultaneously influenced by emissions from mainland China and the Korean Peninsula. In details, as visualized in Figure 4e, the very northern, western, and eastern cruise legs in the Yellow Sea showed relatively low TGM/BC ratios compared to the other cruise periods. These cruise legs above were geographically close to Liaoning province in northeast China, the North China Plain, and the Korean Peninsula. Thus, more influences from the terrestrial emissions explained the low TGM/BC ratios. In contrast, the spatial distribution trend of TGM/BC ratios in the East China Sea was transparent. Overall, we think the application of the TGM/BC ratio is still doable. But the usage of this index should consider various factors as you commented, thus we emphasize that this index is only used as a qualitative index.

In the revision, we have made the following changes.

**Line 401 - 405:**

To evaluate the relative importance of anthropogenic and natural sources to TGM, the ratio of TGM/BC was introduced as a qualitative index. Since TGM and BC shared common anthropogenic sources, and TGM had additional natural sources, an increase in the TGM/BC ratio may indicate the growing importance of natural source contributions, and vice versa.

**Line 410 - 421:**

The cruise measurement illustrated the spatial distribution of the TGM/BC ratio over YS/ECS (Figure 4e). In the East China Sea, the TGM/BC ratio increased with increasing distances away from the coasts. For instance, the TGM/BC ratio near the coasts typically ranged from 0.3 to 5.2 ng µg-1, while offshore values generally fluctuated between 8.6 and 22.9 ng µg-1. This indicated the contribution of natural sources to TGM obviously increased over the open ocean waters. However, this spatial trend was not observed in the Yellow Sea. As depicted in Figure 4e, the very northern, western, and eastern cruise legs in the Yellow Sea showed relatively low TGM/BC ratios compared to the other cruise periods. This phenomenon should be due to the Yellow Sea being a comparatively enclosed basin, as these cruise legs above were geographically close to Liaoning province in northeast China, the North China Plain, and the Korean Peninsula. Thus, more influences from the terrestrial emissions induced the low TGM/BC ratios.

3. Third, I still do not fully understand the authors' choice of the time period for the DSL data. In the original manuscript, I pointed out a mismatch in time periods: DSL data from 2015-2019 and HNI from 10/14/2020-11/4/2020. This was a critical concern, as the authors used the anthropogenic portion of ambient TGM, regressed as a function of BC concentrations from DSL, to further estimate the oceanic flux at HNI. In the revised version, they used DSL data from Oct-Dec 2020, still not a match with the HNI period. Wouldn't the author want to justify their selection of the DSL time frame? Additionally, to separate anthropogenic TGM from TGM in the cruise measurements over the ECS and YS, the authors used the anthropogenic

TGM-BC relationship derived from DSL data when backward trajectories originated from DSL, and used a relationship from JHI when backward trajectories were northerly. However, the JHI period also mismatched the cruise measurement periods. In doing so, the authors seemed to assume that the anthropogenic GEM-BC relationship remains constant over time. If this is indeed their underlying assumption, shouldn't they make an effort to demonstrate its validity?

3.1 In the revised version, they used DSL data from Oct-Dec 2020, still not a match with the HNI period. Wouldn't the author want to justify their selection of the DSL time frame?

Response: Thanks for your comments. In the previous revision, we used DSL data from Oct-Dec 2020 with the purpose of incorporating more data points. As suggested, we have now exactly matched the time period of the DSL data used in our analysis with the observation period at Huaniao Island during October 14 to November 4, 2020. Consequently, Figure 5 in the manuscript has been revised as shown below. Related calculations throughout the text have also been updated accordingly.

3.2 Additionally, to separate anthropogenic TGM from TGM in the cruise measurements over the ECS and YS, the authors used the anthropogenic TGM-BC relationship derived from DSL data when backward trajectories originated from DSL, and used a relationship from JHI when backward trajectories were northerly. However, the JHI period also mismatched the cruise measurement periods. In doing so, the authors seemed to assume that the anthropogenic GEM-BC relationship remains constant over time. If this is indeed their underlying assumption, shouldn't they make an effort to demonstrate its validity?

Response: Thanks for your valuable comments and this is critical for this study. The fact was that due to the incapability of instruments and logistics, the JHI field campaign and the cruise measurements cannot be conducted during the same period. However, the JHI field campaign

and the cruise measurements occurred in the same season in the neighboring years. We think that the anthropogenic TGM-BC relationship should vary little within several years.

In order to validate this, we investigated the relationship between anthropogenic GEM and BC concentrations during winter periods of multiple years before 2020 at Dianshan Lake as shown in the figure below. It can be found that the regression equation (y=0.401x+0.394) during the winter of 2015-2018 was close to the regression equation (y=0.424x+0.171) obtained during this study period (see the revised Figure 5c). However, since there were no long-term measurements of TGM at JHI available, we cannot perform the similar validation as above.

Furthermore, we checked the mercury emission inventory (Feng et al., 2024). China's anthropogenic GEM emissions in 2019 and 2020 were 194.2 tonnes and 191.8 tonnes, respectively, showing negligible changes. Similarly, BC emission inventories in 2019 and 2020 obtained from the Multi-resolution Emission Inventory for China (MEIC, http://meicmodel.org.cn) also exhibited little variation (Geng et al., 2024).

Overall, we think that the relationship between anthropogenic TGM and BC should remain quite constant, especially in neighboring years of 2019 and 2020.

Figure. The relationship between anthropogenic GEM and BC concentrations (daily average) during winter periods of 2015-2018 at Dianshan Lake

In the revision, we have made the following changes.

Line 464 - 468: Furthermore, correlation analysis was conducted between the absolute contribution of anthropogenic sources to GEM and BC, yielding strong correlations at both DSL (Anthropogenic TGM =  $(0.424 \pm 0.014)*BC + (0.171\pm 0.032)$ ,  $R^2 = 0.88$ , Figure 5c) and JHI (Anthropogenic TGM =  $0.362 \pm 0.009)*BC + (0.431 \pm 0.044)$ ,  $R^2 = 0.86$ , Figure 5d), respectively.

Line 472 - 479: To validate the robustness of this relationship in different years, we derived the relationship between anthropogenic GEM and BC at DSL before 2020 based on the same methodology. It can be found that the regression equation during the winter of previous years

was close to that obtained during this study period (Figure S5). In fact, the mercury emissions (Feng et al., 2024) and black carbon emissions (Geng et al., 2024) were quite stable in the neighboring years of this study period. For instance, China's anthropogenic GEM emissions in 2019 and 2020 were 194.2 tonnes and 191.8 tonnes, respectively, showing negligible changes. Thus, it can be assumed that the relationship between anthropogenic GEM and BC remained relatively constant.

4. Fourth, one of my specific comments on the original version was about their explanation for the effect of temperature on seasonal variation in DGM. What they added in lines 557 - 560 merely stated the opposing effects of temperature and didn't really explain how such opposing effects resulted in the seasonal variation their data showed.

Response: Thanks for your comments. The seasonal variation in DGM in this study indicates that the DGM concentrations observed during our winter cruise were significantly lower than those reported in the literature from the summer and autumn cruises in the similar region. The warmer temperature in summer and autumn favored the DGM production in seawater but also promoted the release of DGM from seawater. In principle, the reduction of Hg2+ to DGM and the oxidation of DGM back to Hg2+ in seawater were simultaneously competing processes (O'driscoll et al., 2006). The increased release of DGM due to higher temperature pushes this reaction equilibrium towards reduction, thereby generating more DGM to maintain the high concentrations of DGM in seawater under warmer summer and autumn conditions. As reported extensively in the literature, both dissolved gaseous mercury concentrations and air-sea exchange fluxes were generally higher under warmer conditions (Huang and Zhang, 2021; Kuss et al., 2011; Wang et al., 2019).

In the revised manuscript, we have added the following discussions.

Line 543 - 549: This seasonal variation pattern of seawater DGM, with lower levels in winter compared to summer and autumn, can be attributed to the dynamic equilibrium between competing redox processes. This equilibrium can be represented as:  $Hg^{2+} + photo$ -reductants  $\rightleftharpoons$  DGM + photo-oxidants (O'driscoll et al., 2006). During warmer seasons, higher temperature accelerated the volatilization of DGM from seawater, and also drove the equilibrium toward  $Hg^{2+}$  reduction to replenish the lost DGM. Therefore, DGM concentrations in seawater were usually lower in winter due to suppressed redox processes.

5. Lines 374 - 376: I don't recall Lin et al. (2010b) and Soerensen et al. (2013) suggested that "wetting processes may promote the reduction of Hg to Hg0 in surface seawater". Through what mechanism?

Response: We thank the reviewer for their comments and apologize for the citation error. The reference "Lin et al. (2010b)" should in fact be "Lin et al. (2010a)" from the paper titled Empirical Models for Estimating Mercury Flux from Soils, which reported a positive correlation between Hg flux and soil moisture. Additionally, Soerensen et al. (2013) found that high Hg0 concentrations (leading to high evasion) in near-coastal regions during the August 2008 shelf cruise were attributable to elevated total Hg inputs, partly influenced by rainfall.

However, neither study provides direct evidence supporting the claim that "wetting processes may promote the reduction of Hg(II) to Hg0 in surface seawater." Therefore, in the revised manuscript, we have deleted the sentence "wetting processes may promote the reduction of Hg to Hg0 in surface seawater", and the original phrasing has been revised as follows.

Line 351 - 356: The positive correlation between humidity and TGM may be due to the fact that high humidity is typically associated with the stable atmospheric stratification, which promoted the accumulation of TGM. As for wind speed, it is a key parameter influencing airsea exchange in the double-membrane theory model (Wanninkhof, 1992). For example, Soerensen et al. (2014) found a 2–4 times greater Hg0 flux due to the high wind speed in the Intertropical Convergence Zone (ITCZ) region.

6. Line 499: How was that 5% uncertainty estimated?

Response: When performing the linear regression based on all input data, both the obtained slope and intercept carry uncertainties. The resulting 5% uncertainty ( $\pm 5\%$ ) arises from the propagation of these fitting parameter errors.

7. Lines 588 – 590: I disagree that what they did underscored "the potential effects of diminished anthropogenic emissions on oceanic mercury cycling". See my first comment above.

Response: As shown in the response to the first question, we have removed the related discussions. Thanks for your suggestion.

**References:**

Feng, X., Fu, X., Zhang, H., Wang, X., Jia, L., Zhang, L., Lin, C.-J., Huang, J.-H., Liu, K., and Wang, S.: Combating air pollution significantly reduced air mercury concentrations in China, National Science Review, 11, 10.1093/nsr/nwae264, 2024.

Geng, G. N., Liu, Y. X., Liu, Y., Liu, S. G., Cheng, J., Yan, L., Wu, N. N., Hu, H. W., Tong, D., Zheng, B., Yin, Z. C., He, K. B., and Zhang, Q.: Efficacy of China's clean air actions to tackle PM2.5 pollution between 2013 and 2020, Nat. Geosci., 17, 10.1038/s41561-024-01540-z, 2024.

Huang, S. and Zhang, Y.: Interannual Variability of Air-Sea Exchange of Mercury in the Global Ocean: The "Seesaw Effect" in the Equatorial Pacific and Contributions to the Atmosphere, Environmental science & technology, 55, 7145-7156, 10.1021/acs.est.1c00691, 2021.

Kuss, J., Zülicke, C., Pohl, C., and Schneider, B.: Atlantic mercury emission determined from continuous analysis of the elemental mercury sea-air concentration difference within transects between 50°N and 50°S, 25, 10.1029/2010gb003998, 2011.

O'Driscoll, N. J., Siciliano, S. D., Lean, D. R. S., and Amyot, M.: Gross Photoreduction Kinetics of Mercury in Temperate Freshwater Lakes and Rivers: Application to a General Model of DGM Dynamics, Environmental science & technology, 40, 837-843, 10.1021/es051062y, 2006.

Soerensen, A. L., Mason, R. P., Balcom, P. H., Jacob, D. J., Zhang, Y. X., Kuss, J., and Sunderland, E. M.: Elemental Mercury Concentrations and Fluxes in the Tropical Atmosphere and Ocean, Environmental science & technology, 48, 11312-11319, 10.1021/es503109p, 2014.

Wang, C., Wang, Z., Hui, F., and Zhang, X.: Speciated atmospheric mercury and sea-air exchange of gaseous mercury in the South China Sea, Atmospheric Chemistry and Physics, 19, 10111-10127, 10.5194/acp-19-10111-2019, 2019.

Wanninkhof, R.: RELATIONSHIP BETWEEN WIND-SPEED AND GAS-EXCHANGE OVER THE OCEAN, Journal of Geophysical Research-Oceans, 97, 7373-7382, 10.1029/92jc00188, 1992.